# Distinct kinetics of antibodies to 111 *Plasmodium falciparum* proteins identifies markers of recent malaria exposure

Victor Yman [1,2✉], James Tuju[3], Michael T. White [4], Gathoni Kamuyu[3], Kennedy Mwai[3,5], Nelson Kibinge[3], Muhammad Asghar [1,6], Christopher Sundling [1,6], Klara Sondén[1,6], Linda Murungi[3], Daniel Kiboi[3], Rinter Kimathi[3], Timothy Chege[3], Emily Chepsat[3], Patience Kiyuka[3], Lydia Nyamako[3], Faith H. A. Osier [3,7,8] & Anna Färnert[1,6,8]

Strengthening malaria surveillance is a key intervention needed to reduce the global disease burden. Reliable serological markers of recent malaria exposure could improve current surveillance methods by allowing for accurate estimates of infection incidence from limited data. We studied the IgG antibody response to 111 *Plasmodium falciparum* proteins in 65 adult travellers followed longitudinally after a natural malaria infection in complete absence of re-exposure. We identified a combination of five serological markers that detect exposure within the previous three months with >80% sensitivity and specificity. Using mathematical modelling, we examined the antibody kinetics and determined that responses informative of recent exposure display several distinct characteristics: rapid initial boosting and decay, less inter-individual variation in response kinetics, and minimal persistence over time. Such serological exposure markers could be incorporated into routine malaria surveillance to guide efforts for malaria control and elimination.

[1] Division of Infectious Diseases, Department of Medicine Solna and Center for Molecular Medicine, Karolinska Institutet, Stockholm, Sweden. [2] Department of Infectious Diseases, Södersjukhuset, Stockholm, Sweden. [3] Kenya Medical Research Institute - Wellcome Trust Research Program, Centre for Geographical Medicine Research Coast, Kilifi, Kenya. [4] Department of Parasites and Insect Vectors, Institut Pasteur, Paris, France. [5] Epidemiology and Biostatistics Division, School of Public Health, University of the Witwatersrand, Johannesburg, South Africa. [6] Department of Infectious Diseases, Karolinska, University Hospital, Stockholm, Sweden. [7] Centre for Infectious Diseases, Parasitology, Heidelberg University Hospital, Heidelberg, Germany. [8] These authors contributed equally: Faith H. A. Osier, Anna Färnert. ✉email: victor.yman@ki.se

Reducing the global burden of malaria with the aim of achieving local or regional elimination will require sustained efforts for malaria control[1]. This includes the implementation and the maintenance of high-quality malaria surveillance systems that allow control programs to effectively allocate limited resources in their efforts to reduce disease transmission[2,3].

Serology has been highlighted as a useful complement to traditional methods of surveillance for a wide range of infectious diseases, e.g. dengue fever, trachoma, onchocerciasis, malaria and more recently COVID-19 where it has been evaluated by public health agencies worldwide[4–7]. For malaria, serological surveillance has proven particularly useful in low transmission settings and antibody responses to a number of *Plasmodium falciparum* antigens, from both pre-erythrocytic and blood-stages, have been evaluated as markers of exposure[8–11]. In particular, the responses to merozoite surface protein (MSP) 1 and apical membrane antigen 1 (AMA1) have been found to provide reliable population-level estimates of medium and long-term transmission trends[9,12–14]. However, a serological tool that provides information on the magnitude of the individual-level exposure as well as the time frame within which the individual was last exposed is currently lacking and could improve surveillance by allowing for estimation of infection incidence from single time-point cross-sectional data[15]. Such information could be used to monitor transmission intensity and dynamics, trigger intensified surveillance with focused malaria testing and treatment, guide targeted interventions (e.g. using long-lasting insecticidal nets or other vector control measures) and subsequently evaluate their impact, or even to demonstrate the absence of transmission (reviewed in Greenhouse et al. 2018 and 2019)[16,17].

On the individual level, the magnitude of the malaria-specific antibody response is highly affected by both the time since last infection and the level of prior exposure[18,19]. Although the response is generally considered to be short-lived, accumulating data suggest that the kinetics and the longevity of the response may vary between antigens[18,20–22]. These observations provide a rationale for attempting to identify a combination of antigens to which the antibody responses display distinct kinetics following infection (i.e. some that are short-lived and others that are more long-lived) and allow for accurate estimation of the timing of the individuals last exposure. Ideally, an effective tool for serological surveillance would include only a few antigens in order to be cost-effective and feasible to implement at scale. Identifying the optimal combination of antigens will require a thorough understanding of the kinetics of each candidate antibody response. Given the scarcity of available data on antimalarial antibody kinetics, efforts should preferably start from screening a large number of candidate antigenic targets for suitability[21,23,24].

To date, only a few studies have attempted to identify markers for individual-level exposure, either by analysing cross-sectional data on antibody reactivity in longitudinally monitored individuals in endemic areas[21,25–30] or by analysing longitudinal data on antibody responses obtained from infected individuals participating in controlled human malaria infection (CHMI) trials[31]. Helb et al. used a machine learning approach to identify candidate serological markers of recent infection by analysing cross-sectional data on antibody responses to 655 *P. falciparum* antigens collected at the end of a one-year follow-up of children monitored actively (monthly or three-monthly) and passively for parasitaemia and symptomatic infections, respectively, using microscopic examination of blood slides in an attempt to determine the timing of the last exposure prior to sampling[21]. However, in an endemic setting this approach is notoriously difficult due to undetected exposure and a high frequency of asymptomatic carriage of low-density sub-microscopic infections[32]. Although the timing of exposure can be carefully controlled using CHMI, participants in such trials are typically treated at microscopic or PCR patency of blood-stage infection, i.e. often before symptoms appear[33,34], and the immune response observed may not reflect the response following a symptomatic natural infection[35]. Furthermore, CHMI studies of only primary infections[31] will not capture the effect that repeated parasite exposure may have on antibody profiles and kinetics[19]. It is possible that these uncontrolled factors may have impacted which candidate serological markers have previously been suggested[21,25,26,31].

With the purpose of studying the acquisition and maintenance of both humoral and cell-mediated immunity to malaria, we have established a well-characterised cohort of returning travellers (with different levels of prior malaria exposure) who are followed longitudinally in a malaria free country after successful treatment of a naturally acquired *P. falciparum* infection[19,36–38]. In contrast to the design of the study by Helb et al.[21], samples are collected longitudinally after a known time-point of symptomatic infection. This study design offers a unique opportunity to examine the kinetics of antimalarial immune responses in complete absence of re-exposure. With this near-experimental set-up, we use a recently developed protein microarray (KILchip v1.0[39]) including 111 *P. falciparum* blood-stage antigens to determine the antigen-specificity and kinetics of the antibody response. We identify candidate serological markers of recent malaria exposure and describe how their ability to detect recent exposure depends on the underlying kinetics of each antibody response. We demonstrate that these serological markers are informative also in a moderate transmission setting in Kenya by studying naturally exposed children monitored closely for clinical malaria.

## Results

Sixty-five adults diagnosed with *P. falciparum* malaria at Karolinska University Hospital in Sweden were enroled at the time of diagnosis and followed prospectively with repeated blood sampling (i.e. at enrolment, after approximately ten days, and after one, three, six, and twelve months) for up to one year in complete absence of re-exposure. Out of the 65 participants, 21 were European natives with no prior history of malaria infection who reported a limited time spent in malaria endemic areas and were considered primary infected. The remaining 44 participants (39 born in Sub-Saharan Africa) reported prior malaria episodes, and prolonged residency in malaria endemic areas, and were considered previously exposed (Table 1). Antibody responses to 111 *P. falciparum* blood-stage antigens were quantified in all collected sample series using the KILchip protein microarray. Antibody responses were largely positively correlated (Supplementary Fig. 1) and while many proteins appeared to be highly antigenic only low-level responses were observed towards others (Fig. 1). As expected, the kinetics of the antibody response was antigen-specific but on average the magnitude of the antibody response increased following the acute infection until approximately day 10 (Fig. 2a). After day 10, there was a gradual reduction in the magnitude of the response over time throughout the remainder of the follow-up period. On average, individuals with prior malaria exposure displayed a greater magnitude of the response (Fig. 2a). A similar pattern was observed for the breadth of the response (i.e. the number of antigens to which an individual is seropositive), with the peak in breadth occurring approximately 10 days after the acute infection (primary infected: median = 17, range 7–71; previously exposed: median = 26, range 11–77) (Fig. 2b). Although none of the participants were seropositive for all antigens at any time-point, a majority of previously exposed individuals acquired and maintained a substantially greater breadth of the response at the end of follow-up (primary infected: median = 2, range 0–10; previously exposed: median 3, range 0–42) (Fig. 2b).

**Table 1 Descriptive statistics of the study participants.**

|  | Primary infected | Previously exposed |
|---|---|---|
| Number of participants | 21 | 44 |
| Female sex (%) | 4 (19) | 7 (23) |
| Median age, years (range) | 34 (21–59) | 40 (27–70) |
| Median cumulative time of residency in endemic area, years (range) | 0 (0–3) | 25 (13–39) |
| Median time since residency in endemic area, years (range) | – | 14 (0–46) |
| Median time from symptom onset to diagnosis, days (range) | 3 (0–11) | 3 (1–13) |
| Median parasitaemia, % infected RBCs (range) | 0.45 (<0.1–8.0) | 0.3 (<0.1–7.6) |
| Late treatment failure[a] (%) | 5 (25) | 0 (0) |
| Severe malaria[b] (%) | 1 (5) | 4 (9.7) |
| Treated in intensive care unit (%) | 1 (5) | 2 (4.9) |
| Initial intraveneous artesunate treatment (%) | 6 (30) | 10 (24.4) |

[a]Presented with recrudescent parasitaemia and fever 20–28 days following initial treatment.
[b]Severe malaria was defined according to the WHO criteria which include impaired consciousness, acidosis, hypoglycaemia, severe anaemia, renal impairment, jaundice, pulmonary oedema, bleeding, circulatory shock and hyperparasitemia (Management of Severe Malaria: A Practical Handbook, 3rd edn, 1–83, World Health Organization, 2012)

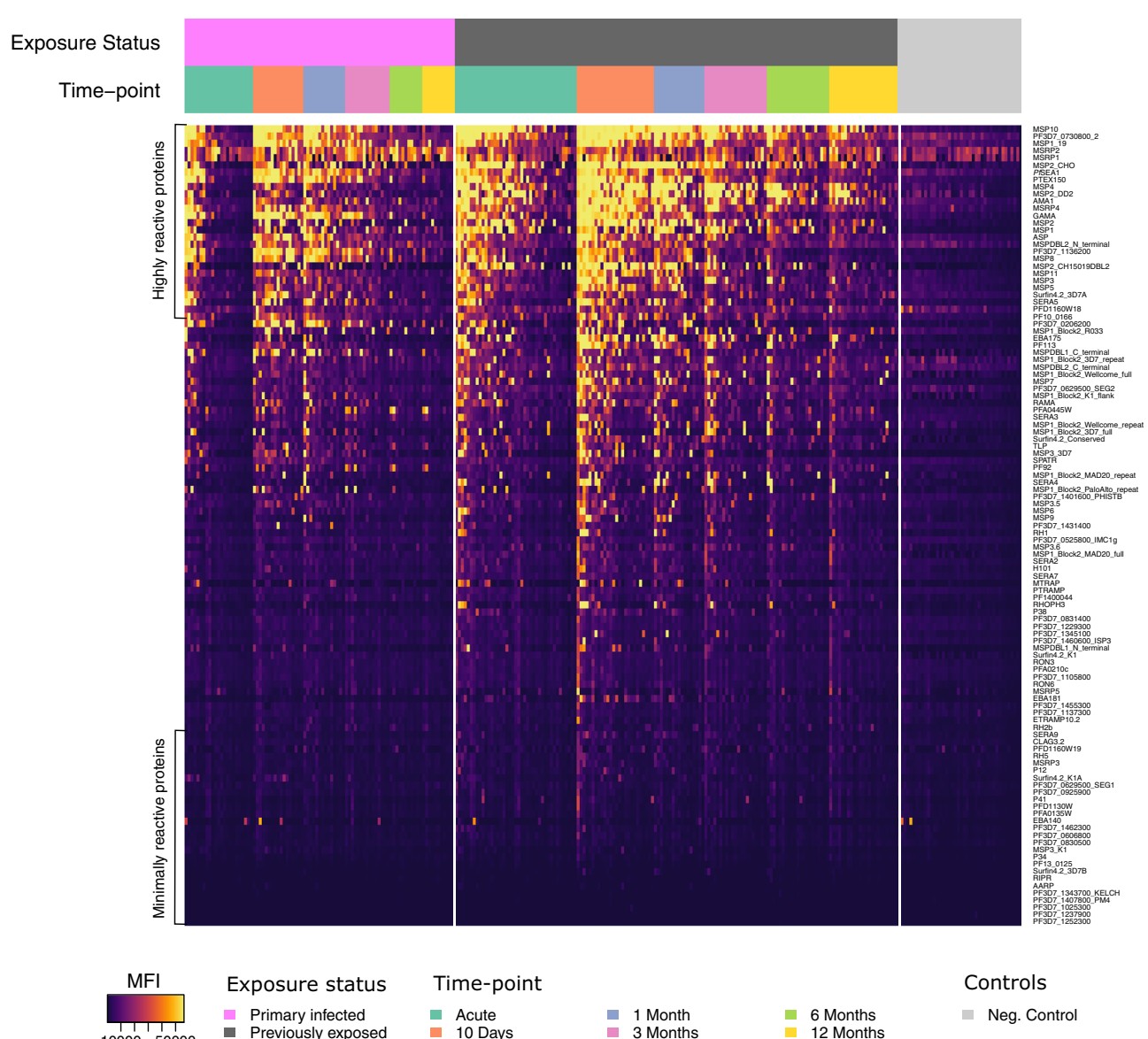

**Fig. 1 A heat map of the normalised median fluorescent intensity (MFI) of the antibody response to each of the 111 antigens included on the KILChip V1.0 Microarray.** Rows correspond to individual antigens while columns correspond to individual samples. Antigens are sorted from top to bottom by decreasing average normalised MFI across all samples. Samples are sorted first by exposure status, second by sampling time-point and third by average normalised MFI across all antigens.

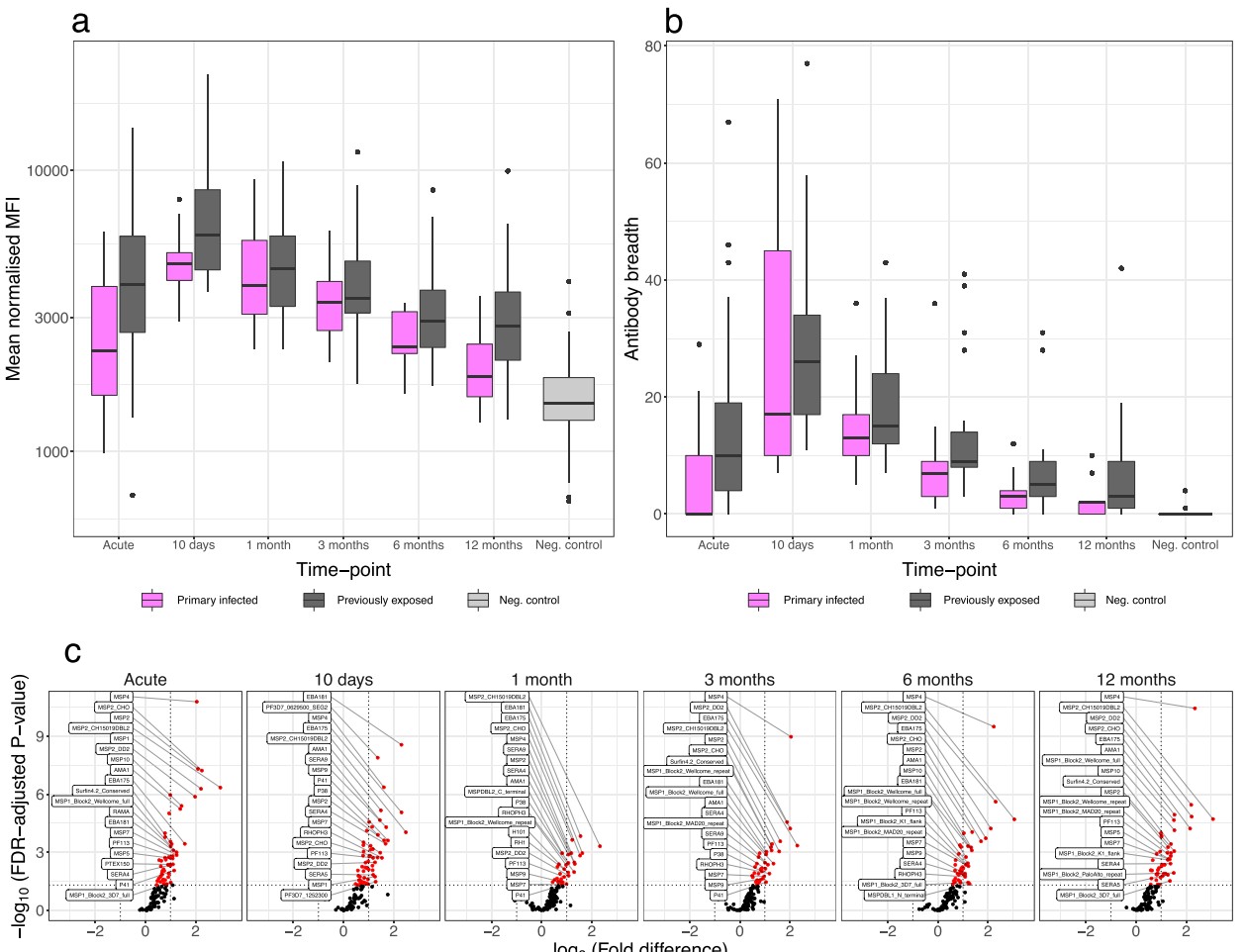

**Fig. 2 Differences in average antibody reactivity related to prior exposure. a** Box-plot of the overall magnitude of the antibody response to *P. falciparum* over time (averaging signal intensities over all antigens for each individual) in individuals with prior malaria exposure (grey; 111 antibody responses (*a*) measured in 149 samples (*s*) from 41 unique individuals (*n*)) or without prior malaria exposure (magenta, $a = 111$, $s = 91$, $n = 24$) and in negative controls (light grey, $a = 111$, $s = 42$, $n = 42$). **b** Box-plot of the breadth of the response over time in individuals with (grey, $a = 111$, $s = 149$, $n = 61$) or without (magenta, $a = 111$, $s = 91$, $n = 24$) prior malaria exposure and in negative controls (light grey, $a = 111$, $s = 42$, $n = 42$). The breadth is expressed as the total number of antigens to which the individual responds. The centres of boxes correspond to the median. The lower and upper hinges of boxes correspond to the first and third quartiles of the data. The upper and lower whiskers extend from the hinges to the largest and smallest values, respectively, no further than 1.5 * the interquartile range from the hinges. Data beyond the end of the whiskers are plotted individually. **c** Volcano plot of the fold-difference in geometric mean antibody reactivity between individuals with or without prior malaria exposure vs. the false discovery rate (FDR)-adjusted *p*-value at each of the sampling time-points. Linear mixed-effect regression models fitted to the Log-transformed antibody data were used to estimate the mean fold-difference of the response for each antigen between primary infected and previously exposed participants. *P*-values were FDR-adjusted for multiple comparisons using the procedure described by Benjamini and Hochberg. A $log_2$ (fold-difference) of greater than 0 indicates antigens to which the geometric mean response is greater among previously exposed individuals and conversely a $log_2$ (fold-difference) of less than 0 indicates antigens to which the geometric mean response is greater among primary infected individuals. Antibody responses that differ significantly between exposure groups are highlighted in red and the 20 antigens for which the difference is greatest are named in the figure. Further details are included within the supplementary information (Supplementary Data 1).

Linear mixed-effect regression models were used to examine differences in the magnitude of the antigen-specific responses between the primary infected and the previously exposed individuals. The previously exposed individuals displayed significantly greater reactivity than the primary infected individuals toward 56 of the 111 antigens at the time of diagnosis, 54 at day 10, 32 at 1 month, 37 at three months, and 44 antigens at both 6 and 12 months of follow-up (Fig. 2c, Supplementary Data 1).

**Individual antibody responses most informative of recent exposure.** What is considered a recent exposure to infection may vary depending on the epidemiological setting and the purpose of

a particular investigation but, in the context of *P. falciparum*, this is often defined as exposure having occurred within the past 3–6 months[17,40]. For the main analysis, samples were treated as independent and a recent exposure was defined as the infection having occurred within 3 months (i.e. 90 days) of sample collection. Consequently, samples collected within 3 months of the acute infection were categorised as obtained from individuals recently exposed to infection whereas the remaining samples were not. This enabled the analysis of a balanced number of samples collected both before (52.5%) and after (47.5%) this temporal threshold within the one-year follow-up (Supplementary Fig. 2). Because a useful serological marker of recent exposure will need to accurately identify recently infected individuals regardless of

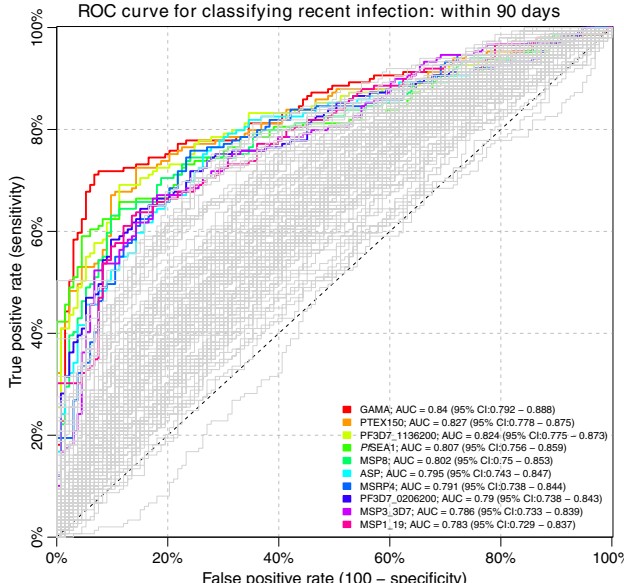

**Fig. 3 Receiver operating characteristic (ROC) curve for classifying individuals as infected within 90 days using a threshold antibody level to a single antigen.** Coloured curves correspond to the top 10 antibody responses that were most accurate in detecting recent infection as determined by the classifier area under the ROC curve (AUC).

their prior level of exposure, data from both exposure groups were analysed jointly. Receiver operating characteristics (ROC) analysis was applied to evaluate whether a threshold level of the antibody response towards a single antigen could be used to accurately classify if a given sample was obtained from a recently exposed individual. The analysis was performed separately for each antibody response and the performance of the classifiers was compared based on the classifier area under the ROC curve (AUC) (Fig. 3). Data on antibody levels towards several individual antigens were able to classify samples as obtained from individuals exposed within the past 3 months with comparable degrees of accuracy (Fig. 3). The best classification performance was obtained using the antibody response towards GPI-anchored micronemal antigen (GAMA) for which the AUC was 0.84 (95% CI: 0.79–0.89) reaching a sensitivity and specificity of 77%. Within this particular cohort this corresponded to an accuracy of 76% and a positive predictive value of 78% and a negative predictive value of 74%. Similar results were obtained using antibody responses towards *Plasmodium* translocon for exported proteins (PTEX) 150, PF3D7_1136200, schizont egress antigen (*Pf*SEA-1), and MSP8 for which the AUCs all exceeded 0.8 (Fig. 3, Supplementary Data 2). In addition, the response towards apical sushi protein (ASP), PF3D7_0206200, MSP7-related protein (MSRP) 4, the 3D7 allelic variant of MSP3, and the 19 kDa fragment of MSP1 (MSP1$_{19}$) were among the top 10 most informative. However, for a majority of responses the classification performance was relatively poor (Fig. 3, Supplementary Data 2).

A sensitivity analysis was performed to examine whether other antibody responses would have been more informative if an alternative definition of recent exposure had been used. The analysis was repeated using several definitions of a recent exposure (i.e. exposure having occurred within 1 month, 2, 3, 4, 6, and 8 months of sample collection). Although, the classifier AUCs varied depending on the definition, the same antibody responses (i.e. GAMA, PTEX150, MSRP4, *Pf*SEA-1, ASP, PF3D7_1136200 and MSP1$_{19}$) were consistently identified among the top 10 responses providing the most accurate identification of recent exposure (Supplementary Fig. 3).

**Combining data on multiple antibody responses.** Combining data on antibody responses towards multiple antigens could theoretically improve the ability to accurately identify recently exposed individuals. Feature selection using a Boruta algorithm was performed to reduce the number of potential combinations to evaluate by selecting only those antibody responses contributing significant information on recent exposure when analysed together for further analysis. It identified 28 antibody responses contributing significant information to the classification of recent exposure (Fig. 4a). Similar to the results based on the threshold antibody level towards a single antigen, the Boruta algorithm identified that the greatest relative importance for classification was contributed by the response towards GAMA, *Pf*SEA1, PF3D7_1136200, PTEX150, and MSP8 (Fig. 4b).

Random forest classifiers were applied to identify a panel of up to five antibody responses informative in identifying recent exposure. The classification performance of all possible two- to five-way combinations of the 28 selected responses was exhaustively evaluated. There was a gradual improvement in classifier performance, i.e. increasing cross-validated AUC, with the sequential increase in panel size from two to five antibody responses. However, each increase in panel size lead to a smaller improvement in classifier performance (Supplementary Fig. 4). The antibody response to GAMA was included in all of the best combinations of two to four antibody responses (Supplementary Fig. 4). The overall best classification performance, with a cross-validated AUC of 0.89 (95% CI: 0.85–0.94) and reaching a sensitivity and specificity of 83%, was obtained for a panel of five antibody responses that included the response to GAMA, MSP1 (full length), both the C- and N-terminal of MSPDBL1, and *Pf*SEA1 (Fig. 4b). This corresponded to an accuracy of 83% and positive and negative predictive values of 84 and 82%, respectively. The responses to GAMA, MSP1 and the N-terminal of MSPDBL1 were included in all of the top 10 most informative panels of size five, and *Pf*SEA1 was included in 8 of the top 10 panels. The classification performance of the top 10 antibody panels was highly comparable with AUCs ranging from 0.88 (95% CI: 0.83–0.94) to 0.89 (95% CI: 0.85–0.94). The random forest classifier based on a combination of five antibody responses provided a substantial improvement in classification accuracy compared to a simple classifier based on a threshold antibody level to GAMA alone. However, no improvement was obtained using a random forest classifier fitted jointly to data on all antibody responses (Cross-validated AUC: 0.83; 95% CI: 0.74–0.89). As an additional evaluation of the robustness of the results obtained using random forest classifiers, the analysis was repeated using logistic regression and yielded results comparable to those obtained using random forests (Supplementary Fig. 5). An alternative approach for cross-validation, which ensures that the same individual is not represented in both the training and the test set, was also evaluated but did not impact the classifier performance (Supplementary Fig. 6).

**Identifying the antibody kinetic properties of a useful serological marker of recent exposure.** Certain antibody responses (e.g. to GAMA and *Pf*SEA-1) were clearly more informative and more useful as serological markers of recent exposure than others, both independently and in combinations including multiple responses. A previously validated antibody kinetic model was applied to quantitatively describe the kinetics of each antibody response to determine if there were underlying kinetic properties shared among informative and non-informative responses. The model, which captures the inter-individual variation in boosting and decay in antibody levels following infection while estimating the average value and variance in the kinetics across the entire

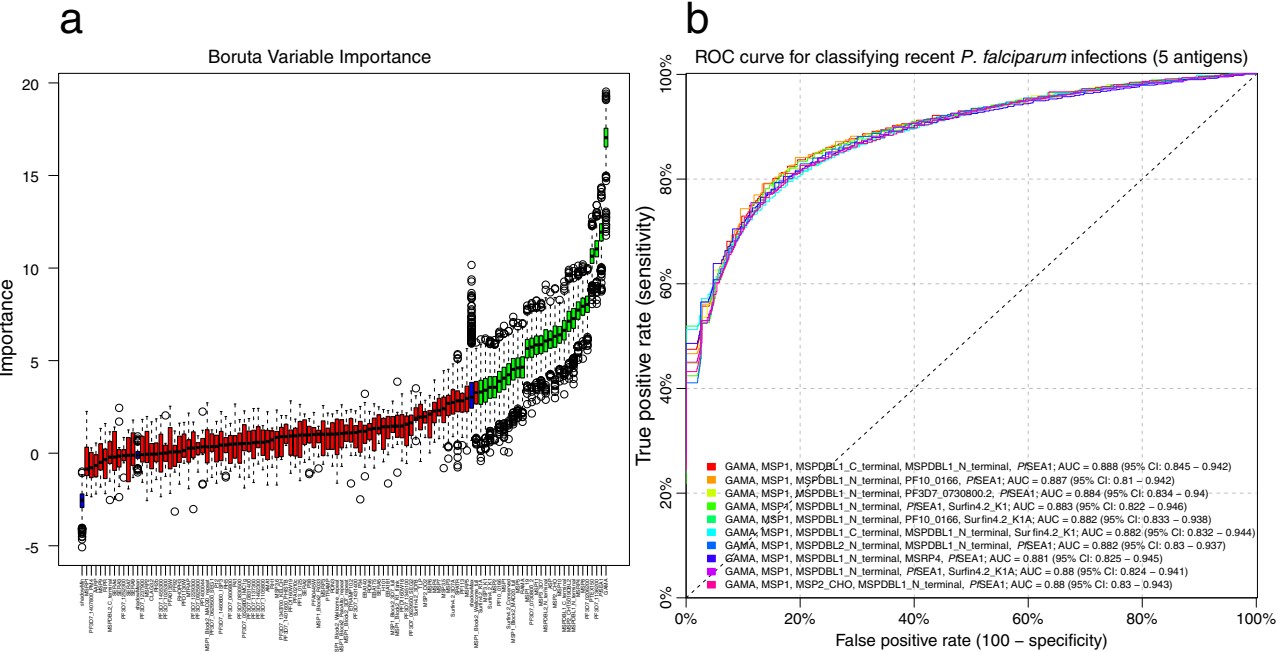

**Fig. 4 Feature selection of antibody responses and evaluation of most informative antibody combinations. a** Variable importance plot for classification performance determined using a Boruta feature selection algorithm as described by Kursa and Rudnicki. The Boruta algorithm was fitted jointly to data for all antibody responses in all samples ($s = 282$) from study participants and controls ($n = 107$). Antibody responses are ordered from left to right by their importance for classification. The importance measure is defined as the Z-score of the mean decrease accuracy (normalised permutation importance). Blue boxes correspond to the minimal, average, and maximum Z-scores of shadow features. Red boxes indicate variables not contributing significantly to accurate classification. Green boxes indicate the 28 antibody responses contributing significantly to classification that were selected for further evaluation. The centres of boxes correspond to the median. The lower and upper hinges of boxes correspond to the first and third quartiles of the data. The upper and lower whiskers extend from the hinges to the largest and smallest values, respectively, no further than 1.5 * the interquartile range from the hinges. Data beyond the end of the whiskers are plotted individually. **b** Cross-validated receiver operating characteristic (ROC) curves. Random forest classifiers fitted to data on antibody responses to the top 10 combinations of 5 out of the 98280 possible combinations of the 28 selected antigens as determined by the classifier area under the ROC curve (AUC). An AUC of 0.5 indicates a classifier that performs no better than random, while an AUC of 1 indicates a perfect classifier. Rainbow coloured lines correspond to the ten classifiers with the highest cross-validated AUCs.

cohort, was fitted separately to data for each antibody response in a Bayesian framework using mixed-effect methods. The model parameters are presented for all antibody responses within the supplementary information (Supplementary Data 3–5). An overview of the different kinetic patterns observed is presented in Fig. 5. The figure includes data and model fits for two representative individuals as well as the model-estimated population-averaged kinetics of the responses towards three antigens, GAMA, EBA175, and PF3D7_1252300, which were identified as highly, moderately, and minimally informative of recent exposure, respectively. The major antibody kinetic patterns observed were: (i) a rapid increase and decay following infection with limited differences between individuals with and without prior exposure (Fig. 5a) (ii) a rapid increase and decay following infection but with substantial differences between individuals with and without prior exposure (Fig. 5b) (iii) a limited boosting and decay following infection with or without differences between individuals with and without prior exposure (Fig. 5c).

To present a meaningful comparison of the different kinetics (i.e. the specific boosting and decay patterns) across all antibody responses, a summary metric of the individual-level antibody kinetics for each participant and antibody response was generated by calculating the relative reduction (%) in antibody levels over the 1-year follow-up. The median relative reduction, as well as the inter-individual variation, differed substantially between antibody responses (Fig. 6, Supplementary Data 6). The greatest relative reductions were estimated for the highly antigenic proteins, e.g. GAMA and PF3D7_1136200, while the smallest relative

reductions were estimated for poorly antigenic proteins, e.g. PF3D7_1343700.KELCH and MSRP5 (Fig. 6). All of the antibody responses that had individually been identified among the top 10 most informative in identifying recent exposure (i.e. GAMA, PTEX150, PF3D7_1136200, *Pf*SEA-1, MSP8, ASP, PF3D7_0206200, MSRP4, MSP3_3D7, MSP1$_{19}$) exhibited a substantial relative reduction in antibody levels during follow-up (Fig. 6). Furthermore, these responses exhibited limited inter-individual variation and limited differences between individuals with different levels of prior malaria exposure and thus a consistent boosting and decay of the response across individuals (Supplementary Fig. 7a, b and Supplementary Fig. 8). For a given antibody response, there was a close association between the estimated relative reduction in antibody levels over time and the performance (AUC) of the corresponding classifier of recent infection (Fig. 7). Multivariable beta-regression models were applied to evaluate the relationship between the relative reduction in antibody levels and the peak antibody level, previous exposure and the number of years the individual had spent in an endemic area. For the majority of antibody responses (81 of 111) the relative reduction in antibody levels was greater if peak antibody reactivity was higher. When accounting for differences in peak antibody levels, the relative reduction in antibody levels was lower in previously exposed individuals for 17 out of 111 antibody responses (Supplementary Data 7). These 17 responses did not include any of the top 10 individually most informative responses. When accounting for differences in both peak antibody levels and previous exposure status there was no significant association between the relative

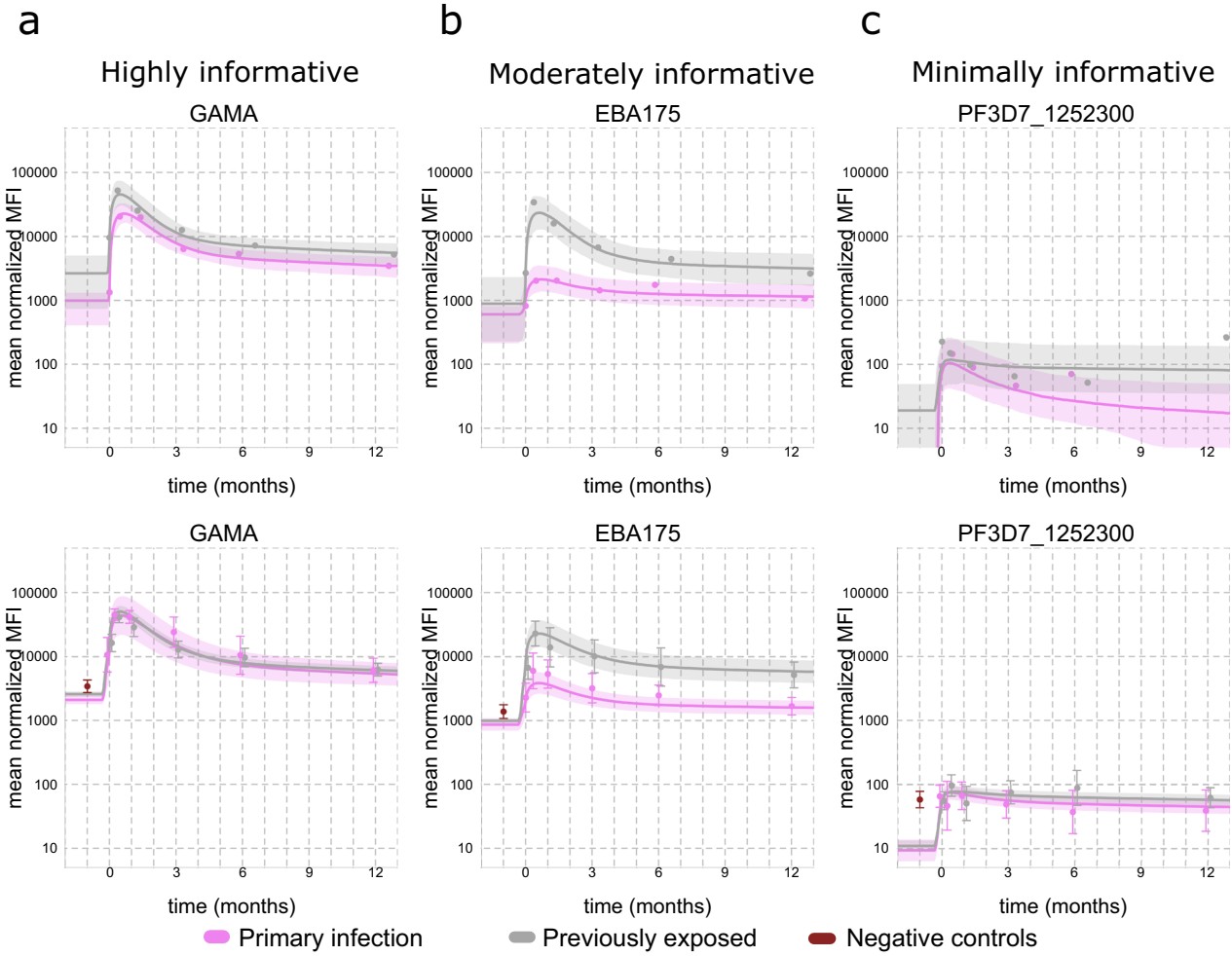

**Fig. 5 Individual- and population-level antibody kinetics for the responses to three representative antigens (GAMA, EBA175, and PF3D7_1252300) in primary infected and previously exposed individuals.** The antibody kinetic model was fitted separately to data on each antibody response measured in 240 longitudinally collected samples from 65 unique study participants. The major antibody kinetic patterns observed were: **a** a rapid increase and decay following infection with limited differences between individuals with and without prior exposure, **b** a rapid increase and decay following infection but with substantial differences between individuals with and without prior exposure, and **c** a limited boosting and decay following infection with or without differences between individuals with or without prior exposure. The top row displays the antibody kinetics for two representative study subjects who were either primary infected or previously exposed. The dots denote the individual sample antibody reactivity, i.e. median fluorescent intensity (MFI). The solid lines denote the model predicted antibody boost and decay patterns relative to the collection of the first sample at time $t = 0$ and the shaded area the 95% credible interval of the prediction. The bottom row displays the geometric mean MFI over time in each exposure group relative to the collection of the first sample at time $t = 0$. The grey and magenta dots denote the average reactivity in previously exposed and primary infected individuals, respectively, at each sampling time point. The red dots denote average reactivity in negative control samples. The error bars of each point denote the corresponding 95% confidence interval (CI). The solid lines denote the model predicted mean boosting and decay in each exposure group and the shaded area the 95% CI.

reduction in antibody levels and the number of years the study participants had spent in an endemic area for any of the measured antibody responses within the travellers cohort (Supplementary Data 7).

**Comparative analysis of antibody response patterns in Kenyan children**. To evaluate the candidate serological markers of recent exposure in an endemic setting, samples from 280 children, age 1–12 years (male: $n = 142$, female $n = 146$), living in a moderate transmission area in Kenya were analysed using the KILchip microarray. Study participants had been monitored continuously for clinical malaria using both passive and weekly active surveillance for 1 year prior to sample collection (Fig. 8a). For the purpose of the analysis, individuals were stratified based on both current infection status and time since last detected clinical episode of malaria (currently infected: $n = 78$, clinical episode within

<3 months: $n = 62$, clinical episode within 3–12 months: $n = 45$, no clinical episode during follow-up: $n = 95$) and by age (<5 years: $n = 114$, 5–12 years: $n = 166$).

Among the Kenyan children, the overall magnitude and the breadth of the response were greatest among individuals who were either currently infected or who had recently had clinical malaria (within 3 months) (Fig. 8b, c). The Kenyan children displayed substantial reactivity to all candidate serological markers of recent exposure identified as individually most informative in adult travellers (Fig. 8d). ROC analysis was applied to evaluate whether a threshold level of the antibody response towards a single antigen could be used to accurately classify if a given sample was obtained from a child who was either currently infected or who had recently had clinical malaria. The best classification performance was obtained for the response towards MSP11 (AUC = 0.81, 95% CI: 0.77–0.86). Similar results were obtained for the responses towards different allelic variants

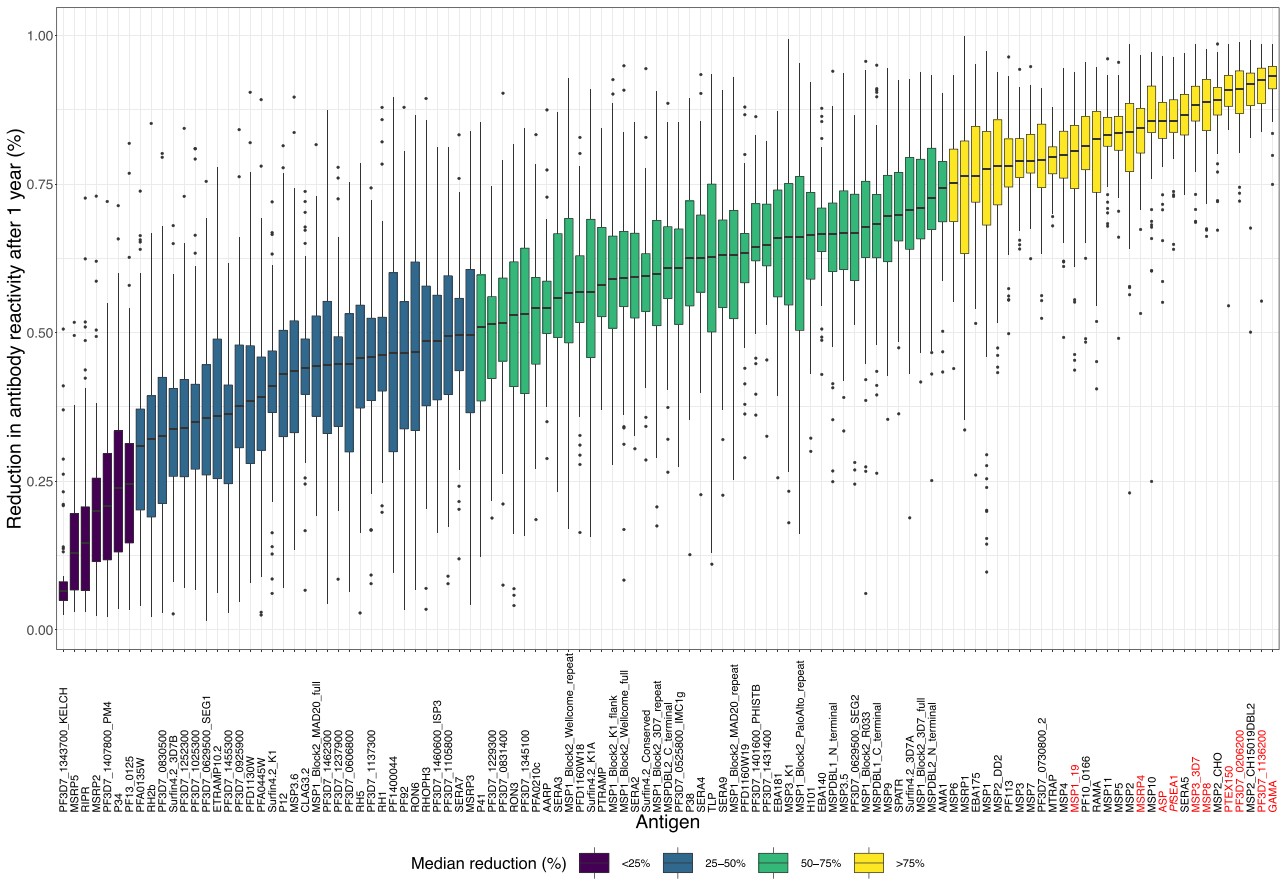

**Fig. 6 Box-plot of individual antigen-specific antibody kinetic model-estimated relative reduction (%) in antibody levels after one year of follow-up.**
The antibody kinetic model was fitted separately to data on each antibody response measured in 240 longitudinally collected samples from 65 unique study participants. Responses are ordered from left to right by smallest to largest relative reduction in antibody levels. The individual responses identified as top 10 most informative in detecting recent infection based on a threshold antibody level to a single antigen are highlighted in red. The colour of the box indicates whether the median relative reduction in antibody levels over one year for a particular response is <25% (purple), 25–50% (petrol), 50–75% (green), or >75% (yellow). The centres of boxes correspond to the median. The lower and upper hinges of boxes correspond to the first and third quartiles of the data. The upper and lower whiskers extend from the hinges to the largest and smallest values, respectively, no further than 1.5 * the interquartile range from the hinges. Data beyond the end of the whiskers are plotted individually.

of MSP2, and towards SERA5, MSP10, MSP4, AMA1, and MSP3.5 for which AUCs ranged from 0.77 (95% CI: 0.72–0.82) to 0.79 (95% CI: 0.75–0.84) (Fig. 9a). Several of these responses, in particular the response towards MSP11 and MSP10, were also identified as highly informative in detecting recent exposure in adult travellers (Supplementary Data 2). The performance of responses identified as individually most informative in adult travellers (i.e. GAMA, PTEX150, PF3D7_1136200, *Pf*SEA1, MSP8, ASP, MSRP4, PF3D7_0206200, MSP3_3D7, and MSP1₁₉) was slightly lower with AUCs ranging from 0.72 (95% CI: 0.68–0.79) to 0.76 (95% CI: 0.71–0.81) (Fig. 9a, Supplementary Data 8). Compared to the travellers, Kenyan children exhibited similar antibody response patterns where levels of antibodies to all candidate serological markers of recent exposure decreased significantly with time since last clinical episode of malaria (Fig. 9b, linear regression model results: Supplementary Data 9). This pattern was consistent in both age groups for all responses except towards *Pf*SEA1 where antibody levels in older children (5–12 years) were stable (Fig. 9b).

## Discussion

Novel and improved tools for malaria transmission surveillance are urgently needed to assist the effective allocation of limited resources for malaria control and assure continued progress

towards malaria elimination[3]. There is a particular need for methods that can detect recent exposure to infection on the individual level which can be used to generate accurate estimates of infection incidence using limited samples and data[15–17]. Here, we screened plasma samples from 65 travellers followed prospectively for up to one year after a naturally acquired *P. falciparum* infection for IgG antibody responses towards 111 blood-stage antigens. Using a data driven approach, we identified candidate serological exposure markers individually informative of recent exposure and demonstrate that combining data on five responses allow for accurate detection of recent exposure to *P. falciparum* within the prior 3-month period. Based on a modelling approach, we then quantitatively examined the kinetics of each individual antibody response and were able to characterise the kinetic properties that make a particular antibody response useful as a serological marker of recent *P. falciparum* exposure. Finally, we demonstrate that the individually informative serological markers of recent exposure can provide information on current infection or recent clinical malaria in naturally exposed children living in a moderate transmission area in Kenya.

When examining each of the 111 antibody responses in travellers individually, we found that the level of the response to several antigens, in particular GAMA, PTEX150, PF3D7_1136200, and *Pf*SEA1, were informative and could be used to identify a

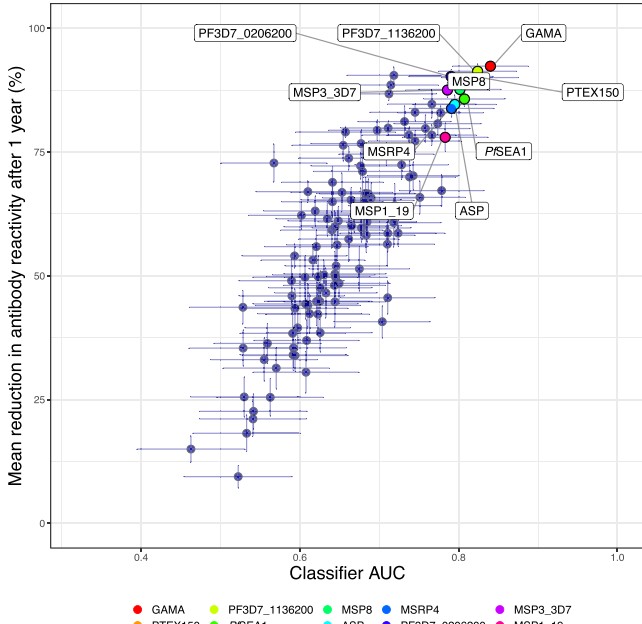

**Fig. 7 Relationship between classification accuracy and the relative reduction in antibody levels.** Antibody kinetic model-estimated mean relative reduction in antibody levels after 1 year of follow-up versus classifier performance as evaluated using the classifier area under the receiver operating characteristic (ROC) curve (AUC). Rainbow colours indicate the antibody responses identified as top 10 most accurate in detecting recent infection, when using the response to a single antigen. Vertical and horizontal error bars represent the 95% confidence interval (CI) of the estimated mean relative reduction in antibody levels and classifier AUCs, respectively. Classifiers were fitted to data on antibody responses measured in 282 samples from 107 unique study participants and controls. Antibody kinetic models were fitted to antibody data measured in 240 samples from 65 unique study participants.

recent exposure with comparable accuracy (Classifier AUCs all exceeding 0.8). The response to GAMA was most informative and it was possible to identify a threshold antibody level such that recently exposed individuals could be identified with a sensitivity and specificity of 77%. The required sensitivity and specificity of a particular surveillance system, and the optimal trade-off between them, should be dictated by the objective of the system, the activity the system is supposed to trigger, the availability of resources and cost of possible interventions[17,41,42]. The level of accuracy in detection of recent exposure achievable using a single antibody response could be acceptable for effective serosurveillance of population-level transmission trends where e.g. a lower sensitivity can be acceptable[43,44].

We demonstrated that the ability to accurately detect recent exposure could be substantially improved if data on up to five antibody responses were analysed simultaneously using a random forest algorithm. We found that the best performance was obtained based on a panel of five antibody responses (AUC = 0.89), reaching a sensitivity and specificity of 83%. There was no single best antibody combination, instead many panels composed of five antibody responses provided comparable results. The existence of many combinations of antibody responses with comparably high accuracy indicates that the superior classification performance of antigen combinations over single antigens is a general phenomenon rather than a chance occurrence. All of the top 10 panels included responses that had individually been identified as highly informative (e.g. to GAMA and *Pf*SEA-1), suggesting that proteins that can

identify recent infections when used individually also do well in combinations. Interestingly, however, they also included responses that were individually not among the more informative (i.e. to MSP1 and either one or both of the N- and C-terminal of MSPDBL1) suggesting that these responses contribute additional information when used in combination with individually informative responses.

The antibody responses to most of the proteins that we identified as informative of recent exposure have to date not been extensively studied. GAMA (85 kDa, 738 amino acids) is a relatively conserved micronemal protein involved in erythrocyte binding and invasion after which the bulk of the protein is shed in soluble form[45]. In addition to expression in blood-stage merozoites GAMA has been reported to be expressed in the micronemes of both salivary gland sporozoites and ookinetes[46,47]. PTEX150 (150 kDa, 993 amino acids) is a conserved protein and one of the core components of the *Plasmodium* translocon for exported proteins responsible for protein trafficking across the parasitophorous vacuole membrane[48]. PF3D7_1136200 (76 kDa, 639 amino acids) is a conserved protein of unknown function to which the antibody response has been associated with protection from clinical disease in cohort studies[49]. *Pf*SEA1 (244 kDa, 2074 amino acids) is a highly invariant vaccine candidate antigen, expressed in late stage schizonts and involved in the egress of the merozoite from the infected erythrocyte, and has been located to the inner leaflet of the red blood cell membrane, the parasitophouros vacuole membrane and maurers clefts[50]. MSP8 (synthesised as an 80 kDa protein, rapidly processed to a 17 kDa fragment, 597 amino acids) is a GPI-anchored protein with limited diversity, predominantly expressed during the trophozoite stage and localised to the parasitophorous vacuole[51]. Among our top 28 candidates, which were informative either individually or in combination, only the responses to *Pf*SEA-1, PTEX150, MSP1 (19 kDa fragment and full length), MSP2 and MSP10 have to our knowledge previously been suggested as markers of recent or concurrent infection[21,22,25,27,31,52]. The response to MSP4 and SERA4 have recently been suggested as markers of recent exposure based on data from primary infections in CHMI trials[31,53]. However, in our study we did not find the response to MSP4 or SERA4 informative in detecting recent exposure in travellers.

It has been suggested that what determines the usefulness of any particular response as a marker of recent exposure is not just the average of its boosting or decay following infection but also the variation in these qualities across individuals[17]. When studied individually, several antibody responses (e.g. to GAMA, PTEX150, PF3D7_1136200, *Pf*SEA-1, and MSP8) were consistently identified as the most informative in detecting recent exposure, suggesting they may share common properties with regards to their kinetics. Because of the longitudinal design of the study, we were able to examine the kinetics of each antibody response in detail using a previously validated mathematical model[18,19]. This allowed us to quantitatively characterise both the antibody boosting and decay, its inter-individual variation as well as its dependency on prior malaria exposure and to identify three key aspects that make a particular antibody response a useful serological marker of recent exposure: (i) a rapid boosting and decay in antibody levels following clearance of infection (ii) limited inter-individual variation in the kinetics (boosting and decay) of the response and therefore predictable kinetics (iii) minimal impact on the kinetics due to prior exposure and a limited formation of an antibody memory response. We could also show that antibody responses that were not informative of recent exposure did not exhibit this behaviour and thereby explicitly demonstrate how the ability to identify recent exposure using serology is based on an understanding of the underlying antibody kinetics.

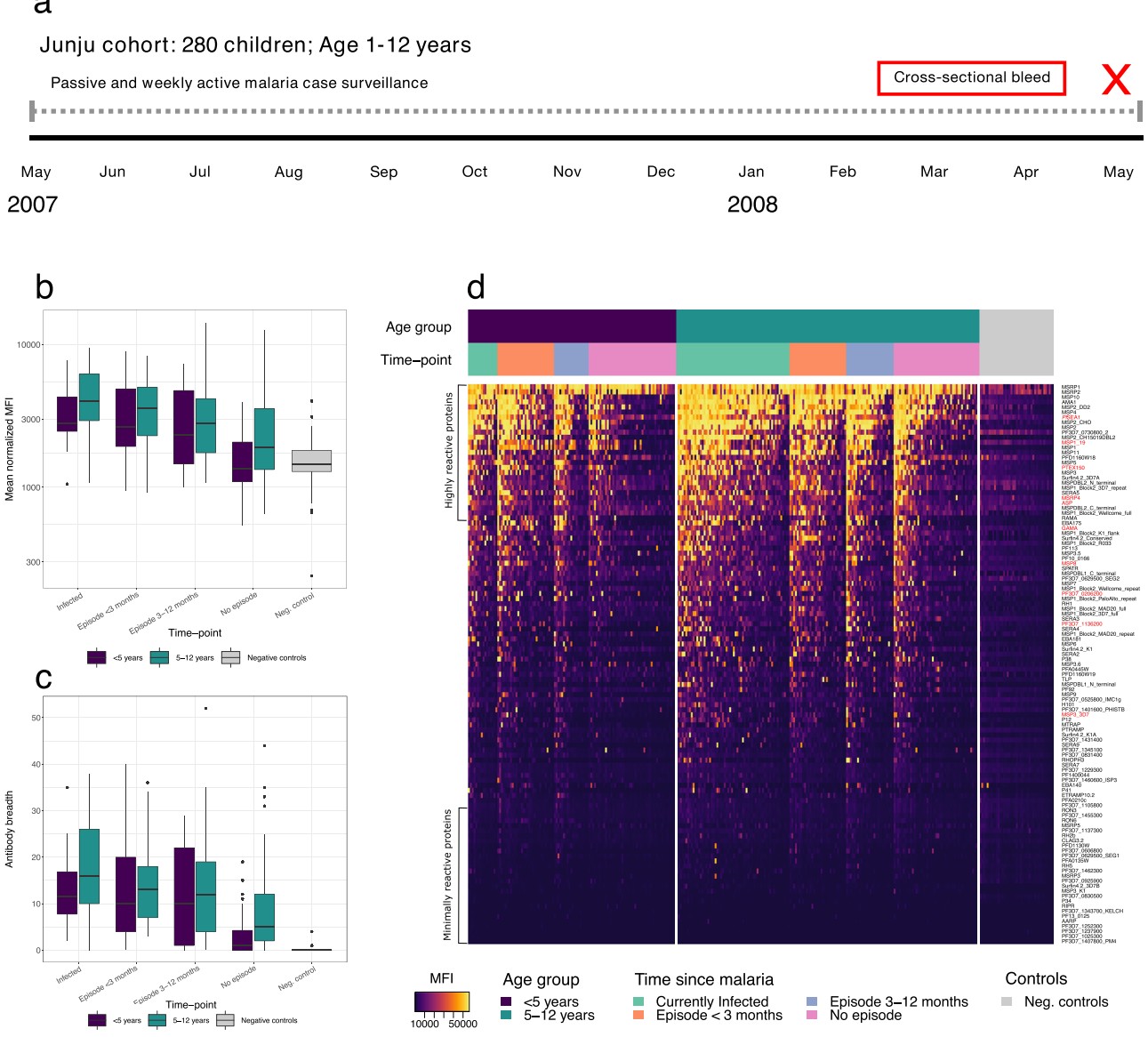

**Fig. 8 Antibody patterns within the Junju cohort. a** Schematic representation of the follow-up and sampling of the Junju cohort. 280 children (age 1–12 years) were monitored for 12 months using passive and weekly active malaria case surveillance. Samples were collected in a cross-sectional bleed at the end of the 12-month follow-up. **b** Box-plot of the overall magnitude of the antibody response to *P. falciparum* relative to current infection status and time since last documented clinical malaria episode (averaging signal intensities over all antigens for each individual) in individuals age <5 years (dark blue, 111 antibody responses (*a*) measured in 114 samples (*s*) from 114 unique study participants (*n*)), age 5–12 years (green, *a* = 111, *s* = 166, *n* = 166) and in negative controls (light grey, *a* = 111, *s* = 42, *n* = 42). **c** Box-plot of the breadth of the response relative to current infection status and time since last documented clinical malaria episode in individuals age <5 years (dark blue, *a* = 111, *s* = 114, *n* = 114), age 5–12 years (green, *a* = 111, *s* = 166, *n* = 166) and in negative controls (light grey, *a* = 111, *s* = 42, *n* = 42). The breadth is expressed as the total number of antigens (out of the 111) to which the individual responds. The centres of boxes correspond to the median. The lower and upper hinges of boxes correspond to the first and third quartiles of the data. The upper and lower whiskers extend from the hinges to the largest and smallest values, respectively, no further than 1.5 * the interquartile range from the hinges. Data beyond the end of the whiskers are plotted individually. **d** A heat map of the normalised median fluorescent intensity (MFI) of the antibody response to each of the 111 antigens within the Junju cohort. Rows correspond to individual antigens while columns correspond to individual samples. Antigens are sorted from top to bottom by decreasing average normalised MFI across all samples. Samples are sorted first by age, second by time since last clinical malaria episode and third by average normalised MFI across all antigens.

The unique longitudinal design of this study, in which the exact time-point of natural exposure is known and where the absence of re-exposure during follow-up can be guaranteed, avoids misclassification of true exposure status thereby limiting bias and providing a unique opportunity to identify markers of recent exposure. Furthermore, including individuals who are both primary infected and previously exposed minimises potential confounding between time since infection and prior exposure

intensity and allowed us to ascertain that our candidate serological markers were able to perform equally well independently of the individuals prior level of exposure.

Although the cohort of travellers serves as an important model population for the discovery of serological exposure markers, it may not be entirely representative of a population living in a malaria endemic setting. The ultimate usefulness of the candidate serological markers as a malaria surveillance tool will depend on

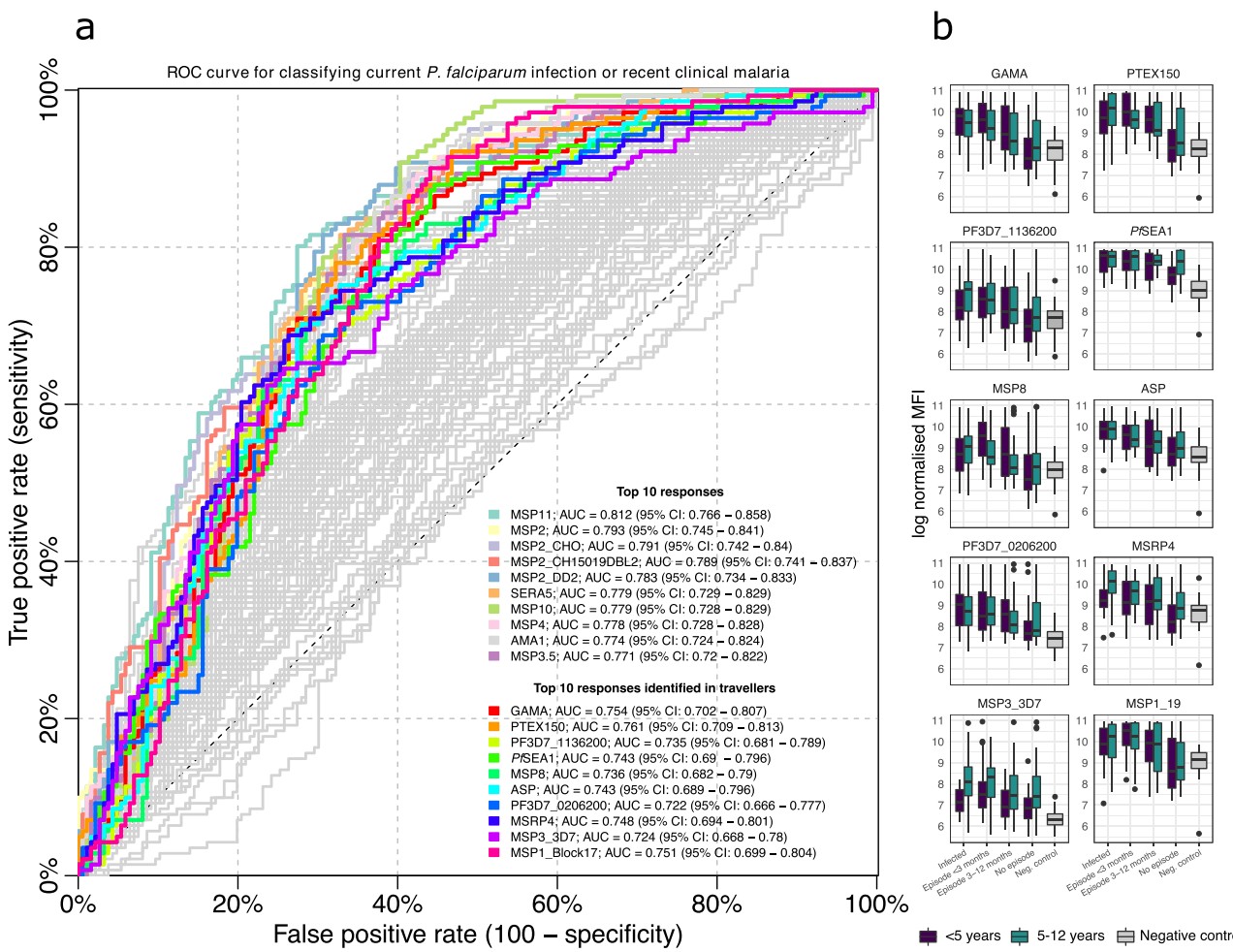

**Fig. 9 Identifying recent infection in Kenyan children. a** Receiver operating characteristic (ROC) curve for classifying individuals as currently infected or having suffered from a clinical malaria episode within 90 days using a threshold antibody level to a single antigen. The first set of coloured curves correspond to the top 10 antibody responses that were most informative as determined by the classifier area under the ROC curve (AUC). The second set of coloured curves indicates the performance of antibody responses identified as individually most informative in detecting recent *P. falciparum* exposure within the longitudinally monitored travellers cohort. **b** Box-plots of the magnitude of the antibody response in Junju children age <5 years (dark blue, 114 samples (*s*) from 114 unique study participants (*n*)) and age 5–12 years (green, *s* = 166, *n* = 166) relative to current infection status and time since last documented clinical malaria episode as well as the reactivity in negative controls (light grey, *s* = 42, *n* = 42) for antibody responses identified as individually most informative in detecting recent *P. falciparum* exposure within the longitudinally monitored travellers cohort. The centres of boxes correspond to the median. The lower and upper hinges of boxes correspond to the first and third quartiles of the data. The upper and lower whiskers extend from the hinges to the largest and smallest values, respectively, no further than 1.5 * the interquartile range from the hinges. Data beyond the end of the whiskers are plotted individually.

their ability to detect recent infection in both adults and children living in endemic settings where re-exposure is common. We therefore also examined the antibody responses towards the top 10 individually most informative candidate serological markers (identified in travellers) in naturally malaria exposed children living in a moderate transmission area in Kenya. We found antibody response patterns comparable to those observed among adult travellers with a decline in antibody levels with time after a symptomatic infection. We found that the level of the response towards individual candidate markers provided information on whether the child was currently infected or had experienced an episode of clinical malaria within the last three months (AUC range: 0.72–0.76). Within the Kenyan cohort antibody responses to MSP11, MSP2, SERA5, MSP10, and MSP4 were most informative in detecting recent symptomatic infection (AUC range: 0.77–0.81). The performance did not differ significantly from the performance of the candidate serological markers identified in adult travellers, for which AUCs were slightly lower. It is

important to note that the results from the travellers and the Kenyan children are not directly comparable due to the fundamentally different study designs and the different types of data analysed which in turn preclude a formal validation of the candidate serological markers of recent infection within the Kenyan cohort. Furthermore, due to the different study designs we do not expect equal performance of the candidate serological markers across both cohorts. In contrast to the travellers, who were sampled longitudinally and in absence of re-exposure after *P. falciparum* infection, the Kenyan children were monitored continuously for one year for clinical malaria, using both passive and weekly active surveillance, and sampled in a cross-sectional bleed at the end of the follow-up period. This design will detect the vast majority of symptomatic *P. falciparum* infections that occur during follow-up but low density and asymptomatic infections will go undetected. It is possible that the antibody responses towards the candidate serological markers of recent infection could have been boosted by this undetected exposure

and that this would have influenced their performance within the Kenyan cohort.

Collectively the results from the travellers and from the Kenyan cohort suggest that the identified candidate responses could be suitable for exposure monitoring in both low and moderate transmission settings[17,54,55]. Additional validation will be required to demonstrate their usefulness, not only in various transmission settings but also across different geographical locations in order to assess the potential impact of parasite genetic diversity on their performance. We aim to pursue this by studying populations from different sites and endemic settings, sampled longitudinally and monitored closely for both symptomatic and asymptomatic *P. falciparum* infections.

In summary, we identify candidate serological markers of recent exposure that, when quantified individually or in combination in a single plasma sample, provide information on when the donor was last exposed to *P. falciparum* infection. Using both a data driven and a modelling approach, we demonstrate that a recent exposure is not necessarily identified by a complex antibody signature that requires sophisticated algorithms for detection but rather by a thorough understanding of the kinetics of the antibody response to a limited number of antigens. We show that the antibody responses towards highly antigenic proteins that demonstrate predictable boosting and decay following infection are sufficient to detect whether a given individual has been exposed within a defined period of time. These candidate serological markers generate information that could be useful for malaria control purposes in order to understand when and where to intensify surveillance, perform targeted testing and treatment, and/or deploy vector control measures, and thereby effectively improve efforts to limit transmission and accelerate progress towards malaria elimination.

## Methods

**Study populations.** The primary study population consisted of adults hospitalised due to *P. falciparum* malaria at the Department of Infectious Diseases at Karolinska University Hospital in Stockholm, Sweden. Study participants were enroled at the time of diagnosis and followed prospectively for up to one year with repeated blood sampling[19]. All participants were treated with a full course of artemether-lumefantrine (AL). Sixteen participants who were vomiting, or who were hyper-parasitaemic (>5% parasitaemia) and/or showing signs of severe malaria (according to the WHO classification[56]) at the time of admission received one to four initial doses of intravenous artesunate followed by a full course of AL. Venous blood samples were collected at the time of enrolment (i.e. at diagnosis) and follow-up samples were collected approximately 10 days, and one, three, six, and twelve months after the first sample. In total, 242 samples were collected from 65 participants. Data on country of birth, previous countries of residence, travel history, use of antimalarial prophylaxis, previous malaria episodes and co-morbidities were collected using a questionnaire administered to each study participant upon enrolment as well as at the end of the follow-up period. Additional clinical data were extracted from hospital records[19].

A secondary study population included 280 children of age 1–12 years enroled in cohort study in Junju village, Kilifi district, Kenya[57]. All children were continuously monitored for clinical malaria using passive and weekly active surveillance for febrile illness for 12 months prior to sample collection (i.e. from May 2007 until May 2008). Symptomatic individuals were tested for parasitaemia using blood smears and all individuals positive for *P. falciparum* were treated for malaria according to Kenyan national guidelines. Samples for serological analysis were collected in a cross-sectional bleed at the beginning of the subsequent more intense malaria transmission season in May 2008[57].

**Ethics statement.** The Swedish study was approved by the Ethical Review Board in Stockholm, Sweden (Dnr 2006/893-31/4 and 2013/550-32/4, 2018/2354-32, 2019-03436) and written informed consent was obtained from all study participants.

The Kenyan study was approved by the Kenya Medical Research Institute (KEMRI) National Ethical Review committee and written informed consent was obtained from the parents and/or guardians of all study participants.

**Protein microarray (KILchip v1.0).** The KILchip v1.0 protein microarray was used for simultaneous quantification of IgG antibody responses to 111 *P. falciparum* antigens[39]. The microarray includes 82 full-length proteins (or for multi-membrane proteins, the largest predicted extracellular loop) and 29 protein

fragments from 8 unique proteins (i.e. MSP1, MSP2, MSP3, MSPDBL1, MSPDBL2, *Pf*SEA-1, PF3D7_06293500 and Surfin 4.2). The proteins were derived from the 3D7 parasite line except for MSP1 Block 2, MSP2, MSP3, and Surfin 4.2 for which five, two, one, and one non-3D7 allelic type(s) were included, respectively. A majority of proteins were produced using a mammalian expression system, while a minority were produced in *Escherichia coli*[39]. Four KILchip v1.0 protein microarray slides were fitted into a hybridisation cassette (Arrayit Corporation ARYC) to obtain a 96-well assay format. After washing four times with 250 μl of HEPES buffered saline (HBS) with 0.1% (v/v) Tween 20 (HBS-Tween) and three times with 250 μl of HBS, 200 μl of blocking buffer, HBS-Tween, with 2% (w/v) bovine serum albumin (BSA) was added to each well and incubated for 2 h at room temperature on a plate shaker. After washing four times, 150 μl of plasma in 1:400 dilution was added to each well and incubated over night at 4 °C on a shaker. After washing, 150 μl of AlexaFluor[647]-Donkey-anti-Human-IgG (Jackson ImmunoResearch, Catalog no.: 709-605-098) was added to each well and incubated for 3 h at room temperature. After final washing, hybridisation cassettes were disassembled, slides rinsed and dried, and then read at 635 nm using a GenePix® 4000B scanner (Molecular Devices) and results obtained using the GenePix® Pro 7 software (Molecular Devices). Positive and negative controls consisting of pooled plasma from malaria exposed Kenyan adults and serum samples from malaria unexposed adult northern European donors without history of travel to malaria endemic countries, respectively, were run on each slide. A 3-fold serially diluted standard calibrator consisting of purified IgG from highly malaria exposed Kenyan donors was assayed once within each batch.

**Data acquisition, cleaning, and normalisation.** R (R: A language and environment for statistical computing, v3.4.4, v3.6.1, and 4.1.1) was used for data processing, normalisation, and analyses. The median fluorescent intensities (MFI) of the local spot background surrounding each spot was subtracted from the MFI of each antigen spot. The mean MFIs of replicate spots were log-transformed to yield an approximate Gaussian distribution of signal intensities. To account for technical slide-to-slide and batch-to-batch variation a two-step normalisation process was applied according to a previously validated procedure[58,59]. First, to account for within batch slide-to-slide effects, a Robust Linear Model (RLM) was fitted to the log-transformed data from the positive control samples assayed on each slide. This was done separately for data from each batch[58]. After obtaining the best-fit parameters for the slide effect the estimated coefficients for each slide was subtracted from all spots within each slide. Following this within-batch RLM normalisation, a second between-batch RLM normalisation was performed similarly using data for the serially diluted standard calibrator. Data for all target antigens that did not demonstrate optical saturation or no signal was used for normalisation in both steps. Following normalisation, the median coefficient of variation (CV) of the antigen-specific batch-to-batch variation was 18.3% (IQR: 15.6–21.5%). A threshold of seropositivity was defined as the mean reactivity + 3 SD of the 42 negative controls. The breadth of the response within each tested sample was defined as the number of antigens for which the reactivity exceeded the seropositivity threshold.

**Evaluating exposure-dependent differences in antibody responses.** Linear mixed-effect regression models were used to identify antigens to which responses were significantly different between primary infected and previously exposed individuals at each sampling time-point. The models were fitted separately to the log-transformed normalised MFI data for each antigen. To account for the false discovery rate (FDR) due to testing such a large number of hypotheses all *p*-values were FDR-adjusted according to the procedures described by Benjamini and Hochberg[60]. FDR-adjusted *p*-values of <0.05 were considered significant.

**Binary classification of recent exposure.** For the purpose of the main analysis a recent exposure was defined as the infection having occurred within 3 months (i.e. 90 days) of sample collection. All samples were categorised as obtained from either a "recently infected" or "not recently infected" individual depending on whether or not they were collected within this specified time frame. To evaluate if the antibody response to any single *P. falciparum* antigen was informative of recent exposure, binary classification using a threshold antibody level was applied to the data for each of the 111 antigens individually using ROC analysis. The AUC was used to compare the classification performance of the individual antibody responses and confidence intervals for the AUCs were estimated using the method described by Sun and Xu[61]. Alternate definitions of recent exposure were also evaluated as part of a sensitivity analysis.

**Feature selection using a Boruta algorithm.** Combining data on multiple antibody responses could theoretically improve the ability to accurately identify recent exposure, however, there are $2^{111}$ potential unique combinations of antibody responses to 111 antigens and to evaluate them all was not feasible[62]. To reduce the number of tentative antibody response combinations to evaluate, feature selection was performed using a Boruta algorithm[63]. The Boruta algorithm is a wrapper method built around a random forest classifier that performs a top-down search for relevant features, while progressively eliminating irrelevant features, by comparing the importance of original features with the importance achievable at random

(estimated using permuted copies of the original features). The algorithm was fitted jointly to antibody data for all 111 antigens.

*Random forest classification based on antibody combinations.* Following feature selection, random forest classifiers were fitted exhaustively to all possible two- to five-way combinations of the down-selected antibody responses in order to evaluate whether a combination of responses could improve the performance of classification of recent infection. Classifier performance was determined by the cross-validated AUC. Cross-validation was performed for each classifier using repeated random sub-sampling by iteratively and randomly splitting the data set into a training set (2/3) and a test set (1/3)[64]. For each split the model was fitted to the training set and the predictive accuracy assessed using the test set. The results from 500 iterations were averaged to obtain a cross-validated estimate of the classifier performance and the 0.025 and 0.975 quantiles of the AUC across iterations were extracted to obtain a 95% confidence interval of the cross-validated AUC.

**Modelling antibody kinetics**. A previously validated mathematical model was used to estimate the antigen-specific antibody kinetics[18,19,28]. The model captures the boosting and bi-phasic decay in antibody levels following infection and quantifies their inter-individual variation, while simultaneously accounting for differences in prior malaria exposure. Briefly the model assumes that the infection causes antibody levels to rise $\tau_0$ days before the individual presents to the hospital (where $\tau_0$ is a parameter estimated for each individual) and that $A(t)$ is the antibody level at time $t > \tau_0$ and is given by the following Eq. (1):

$$A(t) = A_{bg} + A_0 e^{-r_l(t-\tau_0)} + \beta\left((1-\rho)\frac{e^{-r_s(t-\tau_0)} - e^{-r_a(t-\tau_0)}}{r_a - r_s} + \rho\frac{e^{-r_l(t-\tau_0)} - e^{-r_a(t-\tau_0)}}{r_a - r_l}\right) \quad (1)$$

where $r_a$ is the rate of decay of IgG molecules; $r_s$ and $r_l$ are the rates of decay of short- and long-lived antibody secreting cells (ASCs), respectively; $\beta$ is the boost in ASCs following infection at time $\tau_0$; and $\rho$ is the proportion of ASCs that are long-lived. $A_0$ is the pre-existing levels of antibodies. For primary infected individuals, $A_0 = 0$. $A_{bg}$ is the background level of antibody reactivity. The models were fitted separately for each antibody response in a Bayesian framework, and mixed-effect methods were used to capture the natural variation in antibody kinetics between individuals while estimating the average value and variance of the parameters across the entire cohort. Additionally, the antibody kinetic model accounts for sample reactivity exceeding the upper limit of detection of the microarray assay. The rate of decay in antibody reactivity was expressed as the relative reduction (%) after 1 year, starting from the peak of the response[65].

**Association between antibody kinetics and exposure variables**. Multivariable beta-regression models with a logit link function were used to examine the association between antibody kinetic model-estimated relative reduction (%) in antibody reactivity over 1 year and peak antibody reactivity, prior exposure status and years spent in malaria endemic areas. The beta-regression models were used to account for the outcome variable being a rate with values in the standard unit interval (i.e. 0 to 1) and the potential heteroscedasticity and/or skeweness commonly observed with this kind of data and fitted separately to data for each antibody response. To account for the false discovery rate (FDR) due to testing such a large number of hypotheses, all *p*-values were FDR-adjusted according to the procedures described by Benjamini and Hochberg[60]. FDR-adjusted *p*-values of <0.05 were considered significant.

**Antibody levels and time since last malaria episode in Kenyan children**. Linear regression models were used to evaluate the association between the geometric mean antibody response and time since last clinical malaria episode in Kenyan children. The models were fitted separately to the log-transformed normalised MFI data for each antigen. The independent variable, time since last clinical malaria episode, was treated as a categorical variable with the following categories: (i) currently infected, (ii) clinical episode within <3 months, (iii) clinical episode within 3–12 months, (iv) no clinical episode during follow-up. All *p*-values were FDR-adjusted according to the procedures described by Benjamini and Hochberg[60]. FDR-adjusted *p*-values of <0.05 were considered significant.

**Reporting summary**. Further information on research design is available in the Nature Research Reporting Summary linked to this article.

## Data availability
The datasets for the travellers cohort generated and analysed within the current study are included within the supplementary material of this publication (Supplementary Data 10).

## Code availability
The R code and data for reproducing the analysis of the travellers datasets are publicly available under an MIT License online at https://github.com/ymanvictor/Pfalciparum_sero_sign.

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

## Acknowledgements

This work was supported by the Swedish Research Council [grant no 2015-02977 and 2018-02688 to AF] and by the Stockholm County Council [ALF project grant no. 20130207 and 20150135 to AF]. FHAO is supported by a Sofja Kovalevskaja Award from the Alexander von Humboldt Foundation [3.2-1184811-KEN-SKP to FHAO] and an EDCTP Senior Fellowship supported by the European Union [TMA 2015 SF1001 to FHAO]. The funders had no role in study design, data collection and analysis, decision to publish, or preparation of the manuscript. We are grateful to all subjects for their participation in the study. We thank Ingrid Andrén, Irene Nordling and fellow nurses at the Karolinska University Hospital, Department of Infectious Diseases outpatient ward for assistance with coordinating follow-up visits and sampling the study participants. We are grateful to Christine Stenström and colleagues at the Karolinska University Hospital, Department of Microbiology, as well as the attending physicians at the Department of Infectious Diseases, for notifying us regarding the admission of patients diagnosed with *P. falciparum* malaria.

## Author contributions

A.F., F.H.A.O. and V.Y. planned and designed the study. V.Y. organised the enrolment and follow-up of study participants and processed the samples together with K.S., M.A. and C.S., G.K., J.T. and F.H.A.O., designed and developed the protein microarray with assistance from L.M., D.K., R.K., E.C. and P.K., J.T., R.K., T.C. and L.N. performed the microarray experiments. K.M. and N.K. developed the data acquisition pipeline. V.Y. and M.T.W. performed the data analysis, and M.T.W. developed and fitted the antibody kinetic models. V.Y. wrote the first draft of the manuscript. All authors contributed to critically revising the manuscript and have approved the final version.

## Funding

## Competing interests

The authors declare no competing interests.
