## [Peer Review File · Nature Communications]

Distinct kinetics of antibodies to 111 *Plasmodium falciparum* proteins identifies markers of recent malaria exposureREVIEWER COMMENTS

Reviewer #1 (Remarks to the Author):

This paper describes the screening of a larger panel of Pf antigens with samples from returning travellers to calculate kinetics of antibody response to the antigens post infection. The unique elements of the work are the absence of re-exposure in the group which allows the examination of the natural decay in the absence of re-infection. The combination of detailed microarray serology and statistical and modelling analysis make this an extremely interesting and relevant analysis.

Major

1. A concern is the relatively small sample size and the fact that these are drawn from a limited age range and are all adults. Whilst there is no denying the uniqueness of these samples and that the authors could do little to influence it remains a limitation. The lack of children who are a particularly relevant group in terms of public health may well differ in their immune response by virtue of its immaturity and simultaneous challenge with multiple pathogens. I don't think it follows that the same antigenic targets and/or the same kinetics will be identified in this age group. Additionally, it is likely African residents in Europe will be somewhat advantaged socio-economically with fewer concomitant infections. There is little discussion on the implications of not having this group here for the broader. Are data/samples available from Kenya for comparison? These could be from less intensive sampling points but even showing some level of existing reactivity to the prime targets would be informative (see comment about controls below)
2. I would also say that a strength of the study is a limitation in the sense that whilst the clarity of the kinetic signal is due to the absence of reinfection in the study population, reinfection is also the norm. The (in)frequent reinfection may boost differentially and affect kinetics and persistence. Was it possible to disentangle any of this from the data? data in supp table one on sample characteristics (a table I think should be in the main text) show time of and since residency in endemic areas – could these be grouped by likely recent infection (small numbers acknowledged). What about where this residency was? highly endemic areas? east or west Africa? – where any of these factors included in the analysis?
3. I am not sure if I missed this in the methodology or supplementary materials but it would be extremely useful to include more on the antigens. At the moment the paper is peppered with 3 or 4 letter acronyms which are hard to follow for people who are not malaria antigen experts especially given the number of 'novel' targets. It is also key as this may influence rationale choice of antigens if this approach moves further toward diagnostic development. I would suggest some information highlighting some key characteristics – how big are the constructs ie is the response simply due to its size? where is the protein expressed ie is kits reactivity due to its time exposed to the immune system? How is the protein expressed ie is immunogenicity due to expression system? What is the parent strain of the antigen & how much variation is there known to be in the targets ie will this influence geographical response. The authors will have a much better idea of the key components of immunogenicity and could expand this as they see fit
4. I do think the study could be significantly improved by the presentation of data from individuals naturally infected in endemic countries. As in 1) something on the breadth and range of responses in children would be most informative even at a single timepoint (note obviously better). At the moment it is difficult to contextualise the data from the returning travellers.

Minor

Intro

Line 57 – why limit the number of antigens? Presumably any number could be combined either alone or in recombinant chimeric antigens

Line 73 – this statement is really a key issue – undetected exposure and high asymptomatic carriage are the norm in the majority of endemic settings the fact that these confound target identification is just something that needs to be dealt with. It does not automatically follow that targets identified from the current group will be consistent under natural and repeated exposure
Line 78 presumably all infections in this study were treated so this does not seem like a viable argument. It would be helpful to be more balanced in this introduction

Line 98 – as comments above – more info on patients would be helpful and should be prominent. I would suggest the clinical context

Line 108 – why only blood stage? Do authors think other stages would be informative

Line 111 – links to figure 1 it might be helpful to indicate what are considered low responses

Line 153 – this categorisation would be much easier to see either against a level of uninfected controls and/or some assertion of a split in the two phases of the biphasic immune response
Line 178 – addresses the above to a degree but again would have benefitted from more details on negatives and some indication of how repeated measures are dealt with.

Line 422 – a little more on the controls would be helpful ? could these be shown on plots to help orientate readers to range of values in other exposed/non-exposed populations

Line 438 – related to above can the batch to batch variation be described

Line 443 – provide some details on likely range of seropositivity cut offs (again could be included as a line on some box plots)

Line 457 – I read this a couple of times but I could not find if and how repeated measures from the same individual are dealt with or accounted for – I suspect the methods allow for this but it would be important to state this given that repeated observations on relatively few individuals presumably influences outcomes

Reviewer #2 (Remarks to the Author):

This manuscript describes protein microarray experiments performed on longitudinal samples from returned travelers to Sweden with malaria, with the objective of identifying antibodies indicative of recent *P. falciparum* exposure. The study is designed appropriately to answer the question of interest with some forethought given to important details e.g. controls for normalization, and the manuscript is well written and transparent with good attention to methodological detail throughout. The data and results should be of use to the field. I have a few suggestions for improvement, primarily with respect to the interpretation of findings.

Major comments:

1) The study design and returned travelers allows for accurate identification of the amount of time since prior infection, given that additional inoculations will not occur during follow-up in Sweden. However, the fact that most individuals will not have been exposed recently prior to the infection event of interest (since most were not in endemic settings prior to travel) distinguishes this population quite a bit from most individuals living in endemic settings, except for those in very low transmission areas or areas with historically high but recently very low transmission. Thus, while the findings are still of interest, the generalizability to endemic settings is a bit overstated. For example, lines 369 through 373 of the discussion section states that these markers perform well “independently” of prior exposure and thus likely to be suitable for exposure monitoring through a wide range of transmission settings. This will hopefully be the case once tested in endemic settings, but this conclusion is not supported by the current study.

2) A few additional aspects of the study design also preclude broad generalization and should be stated clearly as limitations to generalizability, currently mainly indicated in supplementary table S1. These include a relatively small, mostly male cohort with no children included at all (particularly since children, at least traditionally, are most commonly the targets of malaria surveillance in endemic areas).

3) The authors have done a good job of trying to avoid overfitting by cross validating predictions from the antibody combinations. However, estimates of classification accuracy / sensitivity / specificity are still likely to be overly optimistic even given the caveats in #1 and #2 above, and the authors should either make changes to the methods or at a minimum acknowledge these potential sources of bias. First, the cross validation does not appear to have been performed with sampling by individual, since it is stated that each sample was considered independent. Ensuring that the same individual is not represented in both the training and validation set should be straightforward to perform and the authors are encouraged to do this (or if they have already performed the validation in this fashion this should be clearly stated). Second, cross validation is only performed on the top antigens AFTER these are identified in the entire data set, leading to data leakage. An alternative to produce more accurate estimates of prediction accuracy would be to cross validate the entire process starting with antigen downselection through the final model,

then applying to the validation set. The reviewer recognizes that, given the relatively small sample size and the complexity of communicating these results – they would give e.g. an AUC for a best combination of a certain size but not for a specific combination per se – the authors may not prefer to go this route. However at a minimum this potential bias in estimates of AUC should be acknowledged. Third, the discrete nature of the follow-up time points will inflate estimates of classification accuracy. This is because in a real-world setting people will present with some unknown continuous distribution of time since infection including times near either side of the cutoff which are more likely to be misclassified. For example, in this current study an infection less than 6 months really means <3 months or ≥ 6 months. Measured classification accuracy is thus really comparing these groups with a gap in the middle, which may significantly improve apparent classification accuracy compared to a real target population. The authors cannot change the study design but this source of bias should be acknowledged or the classification should be restated as not e.g. more or less than 6 months but instead distinguishing those with infection more recent than 3 months vs. at least 6 months in the past.

4) While the authors acknowledge increase in performance with multiple antigens was modest, they state that there was an increase in the results section and state that classification accuracy could be “significantly improved” (line 297) with up to five responses versus one in the discussion section. While this may be true, the data do not clearly support this for 2 reasons First, the data shown are for the top 10 AUCs out of 28 (for single targets) vs. out of ~100k (for combinations of 5, thus performance may be expected to appear better even under the null hypothesis of no difference with increasing numbers of antigens number in the combinations just because there are so many more being evaluated and only the top shown. Second, even with this caveat the confidence intervals for top AUCs using 1 vs 5 antigens appear to overlap with point estimates of the other.

5) The authors have provided details of their results in the supplemental tables, which is welcome. The actual data produced by the authors may also be of utility to the scientific community; public release of such datasets has become the standard with such publications and has facilitated advancements. As such, the authors are strongly encouraged to make the normalized (+/- raw) microarray data along with sample metadata directly available with the manuscript as a supplemental data file, or to make these data available in a public repository and not just 'upon reasonable request' from the authors.

Minor comments – authors may feel free to take or leave as they see fit:

6) An additional advantage of the particular study design utilized here is that it minimizes potential confounding between time since infection and prior exposure intensity; the authors may wish to highlight this unique feature of their cohort.

7) The distribution of time since residency in an endemic area (table S1) is only listed as a range of 0 to 42 years. This range is broad and has implications for the interpretability of findings. It would be ideal to present more details on this distribution. If the full data set and metadata are made available (see comment number 4) then this value could e.g. just be listed as a covariate for each individual.

8) In the discussion section, lines 305-308 the authors note the interesting finding that some markers such as MSP1 which may have longer half-lives were selected in combinations. Perhaps such markers may indicate something re: cumulative exposure which improves the ability of the algorithm to better “interpret” the results of shorter-term markers?

9) The kinetics modeling is nice and of interest to provide some intuition for the specific targets which are being selected. Figure 5 is clear but since it only shows 2 individuals it is difficult to get a sense of the distribution across individuals; the bottom row shows the 95% CI for the geo mean not necessarily reflecting the distribution of individual trajectories. It could be useful for the reader to illustrate this directly e.g. as transparent grey lines to show all trajectories or perhaps as a supplemental figure? If the authors provide all of the data (see comment 4) an interested reader could make these plots themselves, though either way this might be a useful way for the authors to communicate biological variability to the broader readership.

10) Some details on convergence of the MCMCs possibly somewhere in the supplemental could be of use.

11) Variable definitions for supplemental tables S4 and S5 may aid in clarity.

12) Very minor: using k-fold cross-validation (e.g. 5 or 10 folds) may improve the efficiency of the process computationally since each observation is guaranteed to be used in the training set and is used in the validation set exactly once, and prediction accuracy as a larger training set is used each time (with $k > 3$) than the CV method currently used.

Reviewer #3 (Remarks to the Author):

Summary: Yman et al., report on the evaluation of human antibody (IgG) responses using a newer microarray chip with human subjects that reside outside endemic areas. A subset of the volunteers have had previous exposures to Pf while another subset were naïve to a primary Pf infection. Human sera were collected acutely and for an extended duration. The results of the analysis identified antibody responses to a set of recombinant proteins/antigens that correlated with a recent infection. The authors propose that the selection of a set of 5 antigens used to measure antibody responses allow for the identification of recent (less than 3 months) infections. It is suggested that this subset of proteins may be used for epidemiological field studies to identify recent infections. The study design is clear, and the results are presented well. Although the broad application of the findings is less clear and requires additional work to justify the claims.

Specific comments:

-Does endemicity impact the assays predictability? In a low endemic area with seasonal transmission (~6 months duration), how does a window of 3 months impact control measures? How do the authors view use of this assay in moderate or high endemicity areas? This reviewer suggests the predictability of the assay should be tested with sera from low and high endemic regions blindly with known infections to establish suitability. There are several longitudinal study sites with suitable information and sera that should permit such an analysis.

-Were the authors able to assess whether the parasite load impacted the antibody titers, kinetics or analyses?

-Fig. 2 legend, do the red circles correspond to only one group or both? This reviewer didn't note whether the distinction was made and suggest it would be helpful.

-The reporting of detection levels is unclear especially for minimal detection which appear to be negative (Fig. 5).

-Of interest, using the kinetic information, could antibody half-lives be determined? If yes, is there a relationship between primary infection and previously exposed individuals for highly or moderately significant antigens?

-Again in Fig. 5, how do the authors explain the variable backgrounds determined for the three proteins included? Are these backgrounds representative?

-Suggest better definition for the same protein that is represented on the microarray using different boundaries e.g., MSP1block17 – is this MSP1-42/19 – if yes then state it.

-Is there any indication of allelic bias in the antibody responses? Does this impact or would this impact field studies?

Line numbers indicated in response to reviewer comments represent line numbers in the revised version of the manuscript with “tracked changes” visible.

REVIEWER COMMENTS

Reviewer #1 (Remarks to the Author):

This paper describes the screening of a larger panel of Pf antigens with samples from returning travellers to calculate kinetics of antibody response to the antigens post infection. The unique elements of the work are the absence of re-exposure in the group which allows the examination of the natural decay in the absence of re-infection. The combination of detailed microarray serology and statistical and modelling analysis make this an extremely interesting and relevant analysis.

Major

1. A concern is the relatively small sample size and the fact that these are drawn from a limited age range and are all adults. Whilst there is no denying the uniqueness of these samples and that the authors could do little to influence it remains a limitation. The lack of children who are a particularly relevant group in terms of public health may well differ in their immune response by virtue of its immaturity and simultaneous challenge with multiple pathogens. I don't think it follows that the same antigenic targets and/or the same kinetics will be identified in this age group. Additionally, it is likely African residents in Europe will be somewhat advantaged socio-economically with fewer concomitant infections. There is little discussion on the implications of not having this group here for the broader. Are data/samples available from Kenya for comparison? These could be from less intensive sampling points but even showing some level of existing reactivity to the prime targets would be informative (see comment about controls below)

This is a valid concern. To address this, we have now analysed samples from 280 children, age 1-12 years, enrolled in cohort study in Junju village, Kilifi district, a moderate transmission area in Kenya. All studied children were continuously monitored for clinical malaria using passive and weekly active surveillance of febrile illness for 12 months prior to sample collection. Symptomatic individuals were tested

for parasitaemia using microscopy of blood smears. Samples for serological analysis were collected in a cross-sectional bleed at the end of the follow-up period. For the purpose of the analysis, individuals were stratified based on current infection status and time since last clinical episode of malaria (Currently infected: n = 78, clinical episode within <3 months: n = 62, clinical episode within 3-12 months: n = 45, no clinical episode during follow-up: n = 95) and by age (<5 years: n = 114, 5-12 years: n = 166).

We found that antibody response patterns were comparable to those observed among adult travellers with a decline in antibody levels with time after symptomatic malaria infection. Furthermore, we found that the level of the response towards individual markers provided information on whether the child was currently infected or had experienced an episode of clinical malaria within the last three months. Although the performance was slightly lower than observed in the travellers (range AUCs 0.72 [95% CI: 0.67-0.77] – 0.76 [95% CI: 0.71-0.81]), it is important to note that the results are not directly comparable due to the fundamentally different study designs and the different types of data analysed. In contrast to the travellers, who were sampled longitudinally and in absence of re-exposure after *P. falciparum* infection, the Kenyan children were monitored continuously for 1 year for clinical malaria, using both passive and weekly active surveillance, and sampled in a cross-sectional bleed at the end of the follow-up period. This design will detect the vast majority of symptomatic *P. falciparum* infections that occur during follow-up but low density and asymptomatic infections will go undetected. It is possible that the candidate serological markers of recent infection could have picked up on this undetected exposure and that this influenced their performance within the Kenyan cohort. Although the differences in study designs between the travellers and the Kenyan children preclude a formal validation of the novel serological markers recent infection within the Kenyan cohort, the results strongly suggest that the candidate markers could be useful in both low and moderate transmission settings.

The new data is presented within the results section lines 289 – 322, within Fig 8 and Fig. 9 as well as Supplementary tables S9 (ROC analysis) and S10 (regression analysis) and discussed on lines 434-458 of the discussion section.

To accommodate the new data within the manuscript word-limit set by the publisher the text within the introduction, results and discussion sections have been shortened. No data has been omitted.

2. I would also say that a strength of the study is a limitation in the sense that whilst the clarity of the kinetic signal is due to the absence of reinfection in the study population, reinfection is also the norm. The (in)frequent reinfection may boost differentially and affect kinetics and persistence. Was it possible to disentangle any of this from the data ? data in supp table one on sample characteristics (a table I think should be in the main text) show time of and since residency in endemic areas – could these be grouped by likely recent infection (small numbers acknowledged). What about where this residency was ? highly endemic areas ? east or west Africa ? – where any of these factors included in the analysis?

These factors were not directly incorporated within the antibody kinetic model structure itself due to the relatively small samples size and the large number of parameters to be estimated by the model.

However, in the revised version of the manuscript we have used multivariable regression models to analyse the association between the antibody kinetic model estimated relative reduction in antibody levels after 1 year (i.e. the antibody kinetic summary metric) and several of the above mentioned factors.

We found that the relative reduction in antibody levels was significantly associated with both the peak antibody levels and with prior exposure status but not with the number of years the individual had spent in an endemic area.

Furthermore, there was no association between either of peak antibody levels or relative reduction in antibody levels and peak parasitaemia. Within the previously exposed group there was no association between years since residency in an endemic area and peak antibody levels or the relative reduction in antibody levels.

Due to small numbers we have not further examined any potential association between boosting and decay patterns and prior country/region of residency and/or current country of travel.

These results have been included within the revised version of the manuscript and are summarised within the results section lines 274 to 288 and are presented in full within the supplementary material (Supplementary Table S7).

As suggested by the reviewer Supplementary Table S1 has been moved to the main body of the manuscript and is now Table 1.

3. I am not sure if I missed this in the methodology or supplementary materials but it would be extremely useful to include more on the antigens. At the moment the paper is peppered with 3 or 4 letter acronyms which are hard to follow for people who are not malaria antigen experts especially given the number of 'novel' targets. It is also key as this may influence rationale choice of antigens if this approach moves further toward diagnostic development. I would suggest some information highlighting some key characteristics – how big are the constructs ie is the response simply due it's size ? where is the protein expressed ie is kits reactivity due to its time exposed to the immune system? How is the protein expressed i.e is immunogenicity due to expression system ? What is the parent strain of the antigen & how much variation is there known to be in the targets i.e will this influence geographical response. The authors will have a much better idea of the key components of immunogenicity and could expand this as they see fit

Additional information regarding the antigenic targets of the antibody responses individually most informative in detecting recent infection has been added to the discussion section (lines 368 - 382). For further details regarding the selection, design and expression of the antigens we refer the reader to the appropriate references detailing the development and validation of the KILchip v 1.0 protein microarray (Kamuyu G et al.; KILchip v1.0: A Novel Plasmodium falciparum Merozoite Protein Microarray to Facilitate Malaria Vaccine Candidate Prioritization. *Front. Immunol.* 2018 and Kamuyu G; Identifying Merozoite Targets of Protective Immunity Against Plasmodium falciparum Malaria; The Open University / KEMRI - Wellcome Trust Research Programme Kenya; 2017).

In the revised version of the manuscript the number of acronyms occurring within the main body of the text have been reduced in order to make the main message of the manuscript easier to follow.

4. I do think the study could be significantly improved by the presentation of data from individuals naturally infected in endemic countries. As in 1) something on the breadth and range of responses in children would be most informative even at a single timepoint (note obviously better). At the moment it is difficult to contextualise the data from the returning travellers.

Please see response to Reviewer #1 Major comment 1 above.

Minor

Intro

Line 57 – why limit the number of antigens? Presumably any number could be combined either alone or in recombinant chimeric antigens

We agree that any number of antigens could theoretically be used, however, in order to be able to implement a serological tool for transmission monitoring at scale, fewer antigens are likely desirable to limit cost of production. Chimeric antigens could be an alternative however antibody responses to different antigens provide different pieces of information regarding the magnitude and timing of prior exposure (e.g. cumulative versus recent exposure) and in our point of view this richer information would be lost if chimeric antigens were to be used. The sentence has been revised to better reflect this (lines 58-59).

Line 73 – this statement is really a key issue – undetected exposure and high asymptomatic carriage are the norm in the majority of endemic settings the fact that these confound target identification is just something that needs to be dealt with. It does not automatically follow that targets identified from the current group will be consistent under natural and repeated exposure

In the revised version of this manuscript we have now included additional antibody data and results from children living in a moderate transmission area in Kenya. Please see the response to Reviewer #1 Major comment 1 for details. In addition, the

implications of undetected exposure when identifying serological markers of recent exposure have been discussed to greater extent (lines 434 - 458).

Line 78 presumably all infections in this study were treated so this does not seem like a viable argument. It would be helpful to be more balanced in this introduction

Following challenge, participants in CHMI studies are continuously monitored for parasitaemia and are treated immediately at microscopic or PCR patency of blood stage infection, i.e. as soon as parasites are detected and in most cases before any symptoms occur. This means that within a CHMI context, participants develop less inflammation and are exposed to parasite antigenic stimulus for a shorter period of time. Their immune response may therefore not fully mirror the response observed during natural symptomatic infection. All of the travellers participating present study suffered from naturally acquired febrile symptomatic *P. falciparum* malaria and presented to the emergency department with substantial parasitemia prior to receiving treatment (see Table 1). The sentence has been revised to clarify this (lines 78 - 80).

Line 98 – as comments above – more info on patients would be helpful and should be prominent. I would suggest the clinical context.

In the revised version of the manuscript Supplementary table S1 has been moved to the main body of the manuscript and is now Table 1. Additional information regarding the study participants has been added to the methods section detailing the study population. See lines 505 – 513.

Line 108 – why only blood stage? Do authors think other stages would be informative

It is entirely possible that antigens from the sporozoite or liver stage could also be informative. However, in the present study we have opted to only examine blood stage targets.

Line 111 – links to figure 1 it might be helpful to indicate what are considered low responses

Figure 1 has been revised. What are considered low-level responses is now indicated.

Line 153 – this categorisation would be much easier to see either against a level of uninfected controls and/or some assertion of a split in the two phases of the biphasic immune response

Data for negative controls have been included in all plots where relevant. Furthermore, a new figure detailing the actual sampling time-points of each participating individual has been included within the supplementary material (Fig S2).

Line 178 – addresses the above to a degree but again would have benefitted from more details on negatives and some indication of how repeated measures are dealt with.

Additional details regarding the negative controls have been added to the methods section (lines 547-548). Furthermore, data from negative controls is now included in all plots where relevant. In addition, the full data set including antibody data and meta data for both travellers and controls has been included within the submission itself (Supplementary File S11).

Mixed effects methods were used to account for the longitudinal structure of the data in all regression models and the antibody kinetic models.

For the classification analysis samples were treated as independent (see line 153) and our primary cross-validation approach was to iteratively randomly select a training data set (2/3 samples) and then validate on the remaining testing data set (1/3 samples) (see lines 618 - 620). There is a theoretical possibility that this may lead to over-fitting, as the training data set will likely contain some samples from ALL individuals.

In the revised version of the manuscript we have examined this risk of overfitting in a sensitivity analysis where the classification analysis has been repeated using an alternative approach to cross-validation, i.e. by selecting a training set of 2/3 individuals (and all their samples) and then validate on a testing set of the remaining

individuals. This alternative approach ensures that the same individual is not represented in both the training and the testing set, thereby substantially reducing the risk of overfitting related to the longitudinal nature of the data. The analysis was repeated using the alternative approach for the top 10 combinations of 5 antibody responses identified using the primary approach.

Using this alternative approach had no impact on the classification performance and the results obtained was more or less identical to the results obtained using the primary approach. This complementary analysis provided no indication overfitting due to the longitudinal nature of the data. In the revised version of the manuscript these new results are presented within Supplementary Fig. S6.

Line 422 – a little more on the controls would be helpful ? could these be shown on plots to help orientate readers to range of values in other exposed/non-exposed populations

Se previous comment.

Line 438 – related to above can the batch to batch variation be described

A quantitative description of batch-to-batch variation is now included within the methods section (see lines 573 - 575)

Line 443 – provide some details on likely range of seropositivity cut offs (again could be included as a line on some box plots)

Data for negative controls have now been included in all plots where relevant. Furthermore the full data set for the travellers including sero-reactivity of both travellers and controls as well as the calculated sero-positivity thresholds have been included as part of the submission (Supplementary file S11).

Line 457 – I read this a couple of times but I could not find if and how repeated measures from the same individual are dealt with or accounted for – I suspect the methods allow for this but it would be important to state this given that repeated

observations on relatively few individuals presumably influences outcomes

Please see above in response to Reviewer #1 minor comment regarding Line 178.

Reviewer #2 (Remarks to the Author):

This manuscript describes protein microarray experiments performed on longitudinal samples from returned travelers to Sweden with malaria, with the objective of identifying antibodies indicative of recent *P. falciparum* exposure. The study is designed appropriately to answer the question of interest with some forethought given to important details e.g. controls for normalization, and the manuscript is well written and transparent with good attention to methodological detail throughout. The data and results should be of use to the field. I have a few suggestions for improvement, primarily with respect to the interpretation of findings.

Major comments:

1) The study design and returned travelers allows for accurate identification of the amount of time since prior infection, given that additional inoculations will not occur during follow-up in Sweden. However, the fact that most individuals will not have been exposed recently prior to the infection event of interest (since most were not in endemic settings prior to travel) distinguishes this population quite a bit from most individuals living in endemic settings, except for those in very low transmission areas or areas with historically high but recently very low transmission. Thus, while the findings are still of interest, the generalizability to endemic settings is a bit overstated. For example, lines 369 through 373 of the discussion section states that these markers perform well "independently" of prior exposure and thus likely to be suitable for exposure monitoring through a wide range of transmission settings. This will hopefully be the case once tested in endemic settings, but this conclusion is not supported by the current study.

In the revised version of this manuscript we have included additional antibody data and results from children living in a moderate transmission area in Kenya in order to evaluate the performance of the candidate serological markers of recent exposure in an endemic setting. Please see response to Reviewer #1 Major comment 1 for details.

The sentence referred to by the reviewer and any related sentences have now been revised to better reflect the conclusions that can be made from the data currently included within this manuscript.

2) A few additional aspects of the study design also preclude broad generalization and should be stated clearly as limitations to generalizability, currently mainly indicated in supplementary table S1. These include a relatively small, mostly male cohort with no children included at all (particularly since children, at least traditionally, are most commonly the targets of malaria surveillance in endemic areas).

Please see previous comment. In addition, Supplementary Table S1 has been moved to the main body of the manuscript and is now Table 1.

3) The authors have done a good job of trying to avoid overfitting by cross validating predictions from the antibody combinations. However, estimates of classification accuracy / sensitivity / specificity are still likely to be overly optimistic even given the caveats in #1 and #2 above, and the authors should either make changes to the methods or at a minimum acknowledge these potential sources of bias.

First, the cross validation does not appear to have been performed with sampling by individual, since it is stated that each sample was considered independent. Ensuring that the same individual is not represented in both the training and validation set should be straightforward to perform and the authors are encouraged to do this (or if they have already performed the validation in this fashion this should be clearly stated).

We have performed this analysis for the 10 best combinations of five responses. The results have been included in supplementary material (Fig. S6). Ensuring that the same individual is not represented in both the training and validation set had no impact on the classification performance as determined by the classifier AUC.

Second, cross validation is only performed on the top antigens AFTER these are identified in the entire data set, leading to data leakage. An alternative to produce

more accurate estimates of prediction accuracy would be to cross validate the entire process starting with antigen downselection through the final model, then applying to the validation set. The reviewer recognizes that, given the relatively small sample size and the complexity of communicating these results – they would give e.g. an AUC for a best combination of a certain size but not for a specific combination per se – the authors may not prefer to go this route. However at a minimum this potential bias in estimates of AUC should be acknowledged.

The reviewer is absolutely correct on this technical point. Our antigen selection was implemented via a two-step process: (i) select optimal antigen combinations (without cross-validation); (ii) evaluate classification performance of optimal antigen combinations (with cross-validation). The cross-validation in the second step ensures that the results that we present are accurate, and not subject to overfitting. The absence of cross-validation in the first step allows for the theoretical possibility that our best antigen combination is not actually the best antigen combination once cross-validation is accounted for. Our reasons for not incorporating cross-validation in the antigen selection process are related to the computational complexity of implementing cross-validation for all combinations. Furthermore, this can be viewed as a conservative stance, as the results we present are robust to cross-validate, but we cannot exclude the possibility that there exist better combinations of antigens.

Third, the discrete nature of the follow-up time points will inflate estimates of classification accuracy. This is because in a real-world setting people will present with some unknown continuous distribution of time since infection including times near either side of the cutoff which are more likely to be misclassified. For example, in this current study an infection less than 6 months really means <3 months or ≥ 6 months.

Measured classification accuracy is thus really comparing these groups with a gap in the middle, which may significantly improve apparent classification accuracy compared to a real target population. The authors cannot change the study design but this source of bias should be acknowledged or the classification should be restated as not e.g. more or less than 6 months but instead distinguishing those with infection more recent than 3 months vs. at least 6 months in the past.

This is an important comment. However, the sampling scheme is not strictly as discrete as it may first appear. Although the aim was to re-sample all subjects at pre-specified follow-up time points after study enrollment (i.e. 10 days, 1, 3, 6, and 12 months) this was not always possible for various reasons (e.g. weekends/holidays, inability of participant to show up for scheduled visits etc.). Instead participants were sampled as close to these pre-specified time-points as possible resulting in a distribution of the number of days since inclusion for all individuals at each pre-specified time point.

In order to more clearly communicate this, a plot detailing the actual sampling time-points of each participating individual has been added to the supplementary material (Fig S2). Furthermore, for completeness all antibody data and meta data including information on when samples were collected has now been included within the submission (Supplementary File S11).

4) While the authors acknowledge increase in performance with multiple antigens was modest, they state that there was an increase in the results section and state that classification accuracy could be "significantly improved" (line 297) with up to five responses versus one in the discussion section. While this may be true, the data do not clearly support this for 2 reasons First, the data shown are for the top 10 AUCs out of 28 (for single targets) vs. out of ~100k (for combinations of 5, thus performance may be expected to appear better even under the null hypothesis of no difference with increasing numbers of antigens number in the combinations just because there are so many more being evaluated and only the top shown. Second, even with this caveat the confidence intervals for top AUCs using 1 vs 5 antigens appear to overlap with point estimates of the other.

The data to support the increase in AUC with increasing numbers of antigens is presented in the cross-validated ROC curves in Figure S2. For example, with 1 antigen AUC = 0.819 (95% CI: 0.749, 0.878), and for 5 antigens AUC = 0.888 (95% CI: 0.845, 0.942). Note that we have not formally tested for the statistical significance of this trend for incremental additions of antigen, but the non-overlap of confidence intervals between 1 antigen and 5 antigens indicates that this is a real trend. As it is not possible to formally test significance (P values etc), we have adjusted our language from "significantly improved" to "substantially improved".

The point about differences in numbers of things being compared: 28 single antigens versus binomial $(28, 5) = 98,280$ combinations of 5 antigens is an interesting one. However, we note focus only on the top 10 combinations (selected by methods without cross-validation as detailed in point 1) rather than all combinations.

5) The authors have provided details of their results in the supplemental tables, which is welcome. The actual data produced by the authors may also be of utility to the scientific community; public release of such datasets has become the standard with such publications and has facilitated advancements. As such, the authors are strongly encouraged to make the normalized (+/- raw) microarray data along with sample metadata directly available with the manuscript as a supplemental data file, or to make these data available in a public repository and not just 'upon reasonable request' from the authors.

The full data set including both antibody data and metadata has now been included within the submission itself (Supplementary File S11).

Minor comments – authors may feel free to take or leave as they see fit:

6) An additional advantage of the particular study design utilized here is that it minimizes potential confounding between time since infection and prior exposure intensity; the authors may wish to highlight this unique feature of their cohort.

We appreciate this comment from the reviewer. An additional sentence referring to this topic has been added (see line 428-432).

7) The distribution of time since residency in an endemic area (table S1) is only listed as a range of 0 to 42 years. This range is broad and has implications for the interpretability of findings. It would be ideal to present more details on this distribution. If the full data set and metadata are made available (see comment number 4) then this value could e.g. just be listed as a covariate for each individual.

The full data set including both antibody data and metadata has now been included within the submission (Supplementary File S11).

8) In the discussion section, lines 305-308 the authors note the interesting finding that some markers such as MSP1 which may have longer half-lives were selected in combinations. Perhaps such markers may indicate something re: cumulative exposure which improves the ability of the algorithm to better “interpret” the results of shorter-term markers?

We agree that this could be a possible explanation although we cannot explicitly determine this using the current data.

9) The kinetics modeling is nice and of interest to provide some intuition for the specific targets which are being selected. Figure 5 is clear but since it only shows 2 individuals it is difficult to get a sense of the distribution across individuals; the bottom row shows the 95% CI for the geo mean not necessarily reflecting the distribution of individual trajectories. It could be useful for the reader to illustrate this directly e.g. as transparent grey lines to show all trajectories or perhaps as a supplemental figure? If the authors provide all of the data (see comment 4) an interested reader could make these plots themselves, though either way this might be a useful way for the authors to communicate biological variability to the broader readership.

We have opted not to include these data in the plots as we feel that the plots would become too crowded and more difficult to interpret. Instead all of the data has now been included as part of this submission (Supplementary File S11) which will enable the interested reader to examine the biological variability in antibody responses across individuals.

10) Some details on convergence of the MCMCs possibly somewhere in the supplemental could be of use.

We did not report MCMC convergence statistics due to the very large number of simulations implemented. The model was implemented separately for each antigen, with each model fit generating chains for all population-level and individual-level parameters. MCMC convergence was assessed via automatic computation of the effective sample size to ensure $ESS > 300$.

11) Variable definitions for supplemental tables S4 and S5 may aid in clarity.

Variable definitions have now been included in all supplemental tables.

12) Very minor: using k-fold cross-validation (e.g. 5 or 10 folds) may improve the efficiency of the process computationally since each observation is guaranteed to be used in the training set and is used in the validation set exactly once, and prediction accuracy as a larger training set is used each time (with $k > 3$) than the CV method currently used.

We appreciate this suggestion from the reviewer. We have evaluated this method, however, this did not impact either efficiency or prediction accuracy.

Reviewer #3 (Remarks to the Author):

Summary: Yman et al., report on the evaluation of human antibody (IgG) responses using a newer microarray chip with human subjects that reside outside endemic areas. A subset of the volunteers have had previous exposures to Pf while another subset were naïve to a primary Pf infection. Human sera were collected acutely and for an extended duration. The results of the analysis identified antibody responses to a set of recombinant proteins/antigens that correlated with a recent infection. The authors propose that the selection of a set of 5 antigens used to measure antibody responses allow for the identification of recent (less than 3 months) infections. It is suggested that this subset of proteins may be used for epidemiological field studies to identify recent infections. The study design is clear, and the results are presented well. Although the broad application of the findings is less clear and requires additional work to justify the claims.

Specific comments:

-Does endemicity impact the assays predictability? In a low endemic area with seasonal transmission (~6 months duration), how does a window of 3 months impact control measures?

We agree that the selection of a 3-month window is somewhat arbitrary and that a different time-frame may be more suitable for monitoring transmission in an area where transmission is strictly seasonal. However, as mentioned within the manuscript lines 150 – 152, this will also depend on the activity the surveillance system is supposed to trigger. A three-month window may still be valid in a strictly seasonal setting if the goal is for example to monitor transmission intensity through cross-sectional survey conducted at the end of the transmission season.

It is because of this we have included the sensitivity analysis within the supplementary material where we demonstrate that it is theoretically possible to use the same serological exposure markers to identify exposure having occurred within 1 month, 2, 3, 4, 6, and 8 months of sample collection (Fig S3).

How do the authors view use of this assay in moderate or high endemicity areas? This reviewer suggests the predictability of the assay should be tested with sera from low and high endemic regions blindly with known infections to establish suitability. There are several longitudinal study sites with suitable information and sera that should permit such an analysis.

This is an important comment. In the revised version of this manuscript we have included additional antibody data and results from children living in a moderate transmission area in Kenya in order to evaluate the performance of the candidate serological markers of recent exposure in an endemic setting. Please see response to Reviewer #1 Major comment 1 for details.

-Were the authors able to assess whether the parasite load impacted the antibody titers, kinetics or analyses?

Please see response to Reviewer #1 Major Comment 2 for details.

Peak parasite load was neither significantly associated peak antibody reactivity nor with the relative reduction in antibody levels over time.

-Fig. 2 legend, do the red circles correspond to only one group or both? This reviewer didn't note whether the distinction was made and suggest it would be helpful.

As explained within the legend of Figure 2C the volcano plots illustrate the relative difference in antibody levels in previously exposed individuals compared to primary infected individuals. Points that fall on the negative end of the x-axis indicate responses that are on average greater in primary infected individuals while points that fall on the positive end of the x-axis indicate responses on average greater among previously exposed individuals.

-The reporting of detection levels is unclear especially for minimal detection which appear to be negative (Fig. 5).

Minimal detection levels are positive. The complete data set analysed, including data for negative controls as well as antigen-specific seropositivity thresholds have now been included within the submission (Supplementary File S11).

-Of interest, using the kinetic information, could antibody half-lives be determined? If yes, is there a relationship between primary infection and previously exposed individuals for highly or moderately significant antigens?

Yes, antibody response half-lives are determined using the antibody kinetic model. The model has been previously described in detail by White et al. JID 2014 and Yman et al. BMC Med. 2019. Briefly, the model assumes that in primary infected individuals, the antigenic stimulus from the malaria infection leads to the proliferation and differentiation of B cells into both short- and long-lived antibody secreting cells (ASCs) that secrete IgG molecules, causing an initial rapid increase in antibody levels. Short- and long-lived ASCs decay at different rates leading to a bi-phasic decay in antibody levels over time, i.e. a short-lived and a long-lived component of the response. The model assumes that individuals with previous exposure may in addition have pre-existing slowly decaying long-lived ASCs generated during previous infections. The model estimates the half-life of secreted antibody molecules and both short- and long-lived ASCs and accounts for exposure-related differences in initial antibody levels, in the magnitude of boosting upon infection, and in the proportion

of short- versus long-lived ASCs. Information regarding both the average (and individual-level) antigen specific half-lives of the short- and long-lived components of the response are available within supplementary Tables S4 and S5 where estimated model population- and individual-level parameters are presented.

Within the manuscript, the overall boosting and decay patterns are summarised as the model estimated relative reduction in antibody reactivity over the 1 year follow-up. In the revised version of the manuscript this data has been included for each individual in supplementary Table S6.

For highly informative antibody responses there was limited difference in response kinetics related to prior exposure status (see lines 269 - 272).

-Again in Fig. 5, how do the authors explain the variable backgrounds determined for the three proteins included? Are these backgrounds representative?

These backgrounds represent the reactivity observed among negative controls. Backgrounds in plots are representative. For the sake of completeness, data for negative controls have now been included in all plots where relevant. Furthermore the complete data set including data for both travellers and negative controls have now been included with the submission.

-Suggest better definition for the same protein that is represented on the microarray using different boundaries e.g., MSP1block17 – is this MSP1-42/19 – if yes then state it.

Yes, this corresponds to the 19 kDa fragment of MSP1 (MSP1₁₉) encoded by Block 17 of the *msp1* gene. All occurrences of this within the manuscript text and figures have been revised to clarify that this refers to MSP1₁₉ and nothing else.

-Is there any indication of allelic bias in the antibody responses? Does this impact or would this impact field studies?

As described within the methods section a majority of the antigens included on the protein microarray were derived from the 3D7 parasite line. However, different major

allelic variants were included for MSP1, MSP2, MSP3 and Surfin4.2. Within both the travellers cohort and the Kenyan cohort the antibody responses to different allelic variants of these proteins were highly positively correlated (see Supplementary Fig. S1) and showed no clear indication of allelic bias.

REVIEWERS' COMMENTS

Reviewer #1 (Remarks to the Author):

Comments on resubmission of Yman

Thank you for giving me the opportunity to re review this paper. I think it has improved significantly and the addition of the analysis of samples collected in Kenya really adds to the paper to the paper.

I do have a couple of further comments which the authors and editors may want to take on board.

1- The discussion does not appear to include much on the limitations of the study. As before and noted by the other reviewer, whilst the returning traveller samples are a unique set they do remain limited in number and in representation. Similarly whilst the inclusion of the Kenya samples are excellent they represent only one endemic site and a relatively Well described one.

2- the language around the use of antigens has improved significantly and is much clearer. However I still think it would be useful to have some more information on the top five or ten selected antigens particularly in relation to their size fragment, isolate used and whether anything is known about antigenic variation or polymorphism. This Could have important bearing on their future use as targets for serological tests.

3- One of the targets MSP-1 has been used in several studies as a marker of cumulative exposure this does not appear to have been referenced or discussed.

Line 136 – Does GAMA include the GPI anchor?

Line 240 – one above – do the authors think the study size is big enough to make this definitive statement

Line 252 – are any samples available for longitudinal assessments from Kenya children? – would be interesting to see

Line 254 onward – I found the text a little confusing around the different categories of time since infection – perhaps the author could re-read – Also unsure as to why you would need serology test for a clinical case.

Line 305 – could sensitivity not be improved rather than settling for lower?

Line 373 – what does frequent mean here?

Line 397 – what are low and moderate settings ?

Reviewer #2 (Remarks to the Author):

The authors are to be commended for their hard work improving what was already a very nicely performed and written manuscript. The inclusion of data from an orthogonal data set of children living in a malaria endemic setting strengthens the conclusions of the work. The performance of the antigens selected in the traveler cohort is somewhat less in this different cohort, which is to be expected and does not lessen enthusiasm for the authors' conclusions to take these candidates forward for more thorough validation, but this expected difference could be acknowledged more directly in the results or discussion.

A few minor comments, primarily for clarity of my prior points:

1) The authors performed an alternative cross-validation as suggested, respecting the independence of samples from particular individuals. This strengthens the results of the model fit (conditional on the variables already having been selected) since results are largely similar, with the only noticeable difference being a drop in sensitivity at the highest specificity level. I believe my original point regarding antigen selection "data leakage" still holds, since all individuals were still used to select the top antigens, with cross validation only serving to make the fit of the model itself on these already selected variables more robust. However, this concern is now mitigated by testing on an orthogonal dataset.

2) I still believe it is difficult to state with any certainty the degree to which multiple antigens (vs 1 or 2) are improving prediction accuracy in the dataset, as the much much larger denominator of models from which only the top are selected makes it almost certain that accuracy would increase, even under the null hypothesis. However this is a minor point.

Reviewer #3 (Remarks to the Author):

The authors' revisions were respectful to the comments made by the reviewers. The initial major finding was a panel of five malaria antigens or serological markers that detected exposure within the previous 3 months with good sensitivity and specificity. As suggested by reviewers, the authors analyzed serum samples from a moderately endemic area. The authors properly discussed the strength and weaknesses of their study design and subsequent analysis. What is challenging, is that even though a panel of serological markers was identified to detect exposure within the previous 3 months, there was a limited overlap with the initial panel. The challenge going forward, to this reviewer based on these findings, is the universal selection of a panel of serological markers. This question is important to discuss to bridge to future efforts or implementation of this approach.

Minor comments: 1) are the authors able to comment on what accounts for the drift in the MSP4 response (Fig. 2C) at 1 month versus the other time points and 2) there are no notations for panel a, b and c in Figure 5.

Line numbers indicated in response to reviewer comments represent line numbers in the revised version of the manuscript with “tracked changes” hidden.

REVIEWERS' COMMENTS

Reviewer #1 (Remarks to the Author):

Comments on resubmission of Yman

Thank you for giving me the opportunity to re review this paper. I think it has improved significantly and the addition of the analysis of samples collected in Kenya really adds to the paper to the paper. I do have a couple of further comments which the authors and editors may want to take on board.

1- The discussion does not appear to include much on the limitations of the study. As before and noted by the other reviewer, whilst the returning traveller samples are a unique set they do remain limited in number and in representation. Similarly whilst the inclusion of the Kenya samples are excellent they represent only one endemic site and a relatively Well described one.

These limitations have now been discussed at greater detail (see lines 379-381 and 412-415)

2- the language around the use of antigens has improved significantly and is much clearer. However I still think it would be useful to have some more information on the top five or ten selected antigens particularly in relation to their size fragment, isolate used and whether anything is known about antigenic variation or polymorphism. This Could have important bearing on their future use as targets for serological tests.

Additional detail regarding the size and antigenic variation has now been provided within in the discussion (lines 327-343). Although all of these 5 proteins appear to be largely conserved, they have to date not been extensively studied and the extent of their antigenic variation has not been described in detail. The proteins on the KILchip v.1.0 were all derived from the 3D7 parasite line (except for protein fragments of MSP1, MSP2, MSP3, and Surfin4.2, for which different allelic variants were included). In the revised version of the manuscript this has been described in the methods section on lines 472-474. An additional comment relating to this topic has been included within the discussion (see lines 412-417).

3- One of the targets MSP-1 has been used in several studies as a marker of cumulative exposure this does not appear to have been referenced or discussed.

This information was included in the initially submitted manuscript but removed in the revised version of the manuscript to be able to comply with the word limit set by the journal. We agree with the reviewer that this is important information to provide context for the present results and realise that it should not have been omitted. It has now been re-added to the manuscript text (see introduction lines 36-38).

Line 136 – Does GAMA include the GPI anchor?

The GAMA protein construct printed on the microarray does not include the GPI anchor. For all proteins included on the array, predicted signal peptides and transmembrane domains were excluded. This is described in detail in the methods paper detailing the development and validation

of the KilCHIP v1.0 protein microarray (Kamuyu et al. Frontriers in Immunology 2018) and which has been cited where appropriate.

Line 240 – one above – do the authors think the study size is big enough to make this definitive statement

The sentence has been revised slightly to clarify that this refers to the present sample set of travellers only (line 249).

Line 252 – are any samples available for longitudinal assessments from Kenya children? – would be interesting to see.

Longitudinal samples from Kenya or other sites are not currently available to us. As discussed on lines 412-417 the candidate serological markers will have to be validated using longitudinally sampled cohorts in different geographical regions and transmission settings. We aim to pursue this in future studies.

Line 254 onward – I found the text a little confusing around the different categories of time since infection – perhaps the author could re-read – Also unsure as to why you would need serology test for a clinical case.

The text has been revised slightly. One would not need a serological test to identify a clinical case per se. However, as described within lines 249-255 (and in more detail within the methods section) the Kenyan children were monitored for symptomatic infections (clinical disease) only and asymptomatic infections would have gone undetected. Therefore, we could only evaluate the performance of the serological markers for detecting recent clinical malaria and not any type of exposure. Furthermore, we wanted to impose a categorization on the samples from the Kenyan children that was as similar as possible to the categorization used for the travellers' samples.

Line 305 – could sensitivity not be improved rather than settling for lower?

Yes, this is of course the goal and the reason why we evaluate different combinations of serological markers, i.e. to improve both sensitivity and specificity in detecting recent exposure. Having said that, it may still be sufficient with a lower sensitivity if the purpose is to evaluate population level trends in transmission intensity (as has previously been done; see discussion lines 307-309 and references 40 and 41) rather than identifying individual-level recent exposure, as stated within the manuscript text.

Line 373 – what does frequent mean here?

In many endemic settings individuals will suffer multiple malaria infections each year. As expected several children in the Kenyan cohort had multiple clinical episodes of malaria during the year of follow-up. The frequency of repeated infections will, among other things, depend on the local transmission setting. The sentence has been revised slightly by substituting the word “frequent” for “common”.

Line 397 – what are low and moderate settings ?

This refers to low and moderate malaria transmission settings as traditionally defined by the point prevalence of infection in children age 2-9 years. A reference has been included to provide background for this classification (Hay S I et al. Lancet Inf. Dis. 2008).

Reviewer #2 (Remarks to the Author):

The authors are to be commended for their hard work improving what was already a very nicely performed and written manuscript. The inclusion of data from an orthogonal data set of children living in a malaria endemic setting strengthens the conclusions of the work. The performance of the antigens selected in the traveler cohort is somewhat less in this different cohort, which is to be expected and does not lessen enthusiasm for the authors' conclusions to take these candidates forward for more thorough validation, but this expected difference could be acknowledged more directly in the results or discussion.

We appreciate this suggestion by the reviewer. The differences have now been acknowledged more directly in the both the results (see lines 272-273) and discussion sections (see lines 393-394)

A few minor comments, primarily for clarity of my prior points:

1) The authors performed an alternative cross-validation as suggested, respecting the independence of samples from particular individuals. This strengthens the results of the model fit (conditional on the variables already having been selected) since results are largely similar, with the only noticeable difference being a drop in sensitivity at the highest specificity level. I believe my original point regarding antigen selection "data leakage" still holds, since all individuals were still used to select the top antigens, with cross validation only serving to make the fit of the model itself on these already selected variables more robust. However, this concern is now mitigated by testing on an orthogonal dataset.

2) I still believe it is difficult to state with any certainty the degree to which multiple antigens (vs 1 or 2) are improving prediction accuracy in the dataset, as the much much larger denominator of models from which only the top are selected makes it almost certain that accuracy would increase, even under the null hypothesis. However this is a minor point.

The existence of many antibody response combinations with comparably high accuracy indicates that the superior classification performance of antigen combinations over single antigens is a general phenomenon rather than a chance occurrence. The text in discussion (lines 316-319) has been updated to reflect this point. Furthermore, the accuracy of the tested combinations has been demonstrated to be valid after cross-validation.

Reviewer #3 (Remarks to the Author):

The authors' revisions were respectful to the comments made by the reviewers. The initial major finding was a panel of five malaria antigens or serological markers that detected exposure within the previous 3 months with good sensitivity and specificity. As suggested by reviewers, the authors analyzed serum samples from a moderately endemic area. The authors properly discussed the strength and weaknesses of their study design and subsequent analysis.

What is challenging, is that even though a panel of serological markers was identified to detect exposure within the previous 3 months, there was a limited overlap with the initial panel.

The challenge going forward, to this reviewer based on these findings, is the universal selection of a panel of serological markers. This question is important to discuss to bridge to future efforts or implementation of this approach.

The performance of the top 10 individually most informative markers of recent exposure was slightly lower in the Kenyan cohort than observed in the travellers (range AUCs 0.72 [95% CI: 0.67-0.77] – 0.76 [95% CI: 0.71-0.81]). This is to be expected and results are not directly comparable across cohorts due to the fundamentally different study designs and the different types of data analysed. In contrast to the travellers, who were sampled longitudinally and in absence of re-exposure after *P. falciparum* infection, the Kenyan children were monitored continuously for 1 year for clinical malaria, using both passive and weekly active surveillance, and sampled in a cross-sectional bleed at the end of the follow-up period. This design will detect the vast majority of symptomatic *P. falciparum* infections that occur during follow-up but low density and asymptomatic infections will go undetected. It is possible that the candidate serological markers of recent infection could have picked up on this undetected exposure and that this influenced their performance within the Kenyan cohort. Although the differences in study designs between the travellers and the Kenyan children preclude a formal validation of the novel serological markers recent infection within the Kenyan cohort, the results strongly suggest that the candidate markers could be useful in both low and moderate transmission settings. This is described with additional detail within the discussion (lines 388-408)

Furthermore, when examining the performance of different combinations of antibody responses for detecting recent exposure within the travellers cohort, we found that there was no single best antibody combination, instead many panels composed of five antibody responses provided comparable results. Although it may not be possible to identify a universally useful combination of responses our results indicate that it could be possible to identify a combination of responses that is at least widely applicable. As discussed on lines 412-417 this will require additional validation studies prior to attempting implementation of the candidate serological markers of recent exposure for surveillance purposes.

Minor comments:

1) are the authors able to comment on what accounts for the drift in the MSP4 response (Fig. 2C) at 1 month versus the other time points and

We are a somewhat unsure of what the reviewer refers to by a “drift in the MSP4 response”. The relative difference (fold-difference) between exposure groups is somewhat smaller at one month compared to the other time-points this could partly be explained by a larger proportion of samples within the previously exposed group reaching the upper limit of detection of the microarray assay at around 1 month when antibody responses are at their peak. This gives the impression that the difference is smaller (full data available within supplementary data 1). The antibody kinetics model accounts for the fact that the reactivity of some samples exceed the upper limit of detection of the assay and allows us to model differences in kinetics between groups during the entire follow-up period (data on differences in peak levels available in supplementary data 5).

2) there are no notations for panel a, b and c in Figure 5.

Panel labels have now been added.